# PAMPs and DAMPs in Sepsis: A Review of Their Molecular Features and Potential Clinical Implications

**DOI:** 10.3390/ijms25020962

**Published:** 2024-01-12

**Authors:** Sara Cicchinelli, Giulia Pignataro, Stefania Gemma, Andrea Piccioni, Domitilla Picozzi, Veronica Ojetti, Francesco Franceschi, Marcello Candelli

**Affiliations:** 1Department of Emergency, S.S. Filippo e Nicola Hospital, 67051 Avezzano, Italy; cicchinelli.sara90@gmail.com; 2Department of Emergency, Anesthesiological and Reanimation Sciences, Fondazione Policlinico Universitario Agostino Gemelli—IRRCS, Università Cattolica del Sacro Cuore, 00168 Roma, Italy; giulia.pignataro@policlinicogemelli.it (G.P.); stefania.gemma01@icatt.it (S.G.); andrea.piccioni@policlinicogemelli.it (A.P.); domitilla.picozzi01@icatt.it (D.P.); veronica.ojetti@policlinicogemelli.it (V.O.); francesco.franceschi@policlinicogemelli.it (F.F.)

**Keywords:** DAMPs, PAMPs, innate immunity, sepsis, septic shock

## Abstract

Sepsis is a serious organ dysfunction caused by a dysregulated immune host reaction to a pathogen. The innate immunity is programmed to react immediately to conserved molecules, released by the pathogens (PAMPs), and the host (DAMPs). We aimed to review the molecular mechanisms of the early phases of sepsis, focusing on PAMPs, DAMPs, and their related pathways, to identify potential biomarkers. We included studies published in English and searched on PubMed^®^ and Cochrane^®^. After a detailed discussion on the actual knowledge of PAMPs/DAMPs, we analyzed their role in the different organs affected by sepsis, trying to elucidate the molecular basis of some of the most-used prognostic scores for sepsis. Furthermore, we described a chronological trend for the release of PAMPs/DAMPs that may be useful to identify different subsets of septic patients, who may benefit from targeted therapies. These findings are preliminary since these pathways seem to be strongly influenced by the peculiar characteristics of different pathogens and host features. Due to these reasons, while initial findings are promising, additional studies are necessary to clarify the potential involvement of these molecular patterns in the natural evolution of sepsis and to facilitate their transition into the clinical setting.

## 1. Introduction

Sepsis is a serious organ dysfunction caused by a dysregulated host reaction to an infective insult. It has a large impact on the public healthcare system because of its high incidence, mortality, and treatment costs. In fact, it represents the first cause of decease from infectious diseases, especially if not diagnosed and cured promptly [1]. In 2017, 48.9 million cases were registered worldwide, with about 11.0 million sepsis-related deaths, representing 19.7% of all global deaths. Even though age-standardized incidence fell by 37.0% and mortality decreased by 52.8% from 1990 to 2017, sepsis remains a massive cause of health loss worldwide. In the USA, it represents the most common cause of in-hospital deaths and costs more than USD 24 billion/year. Sepsis most commonly affects females and occurs in individuals with underlying causes of health loss, like previous infectious disease, trauma, stroke, cirrhosis, diabetes, heart failure, chronic kidney diseases, and others. However, it does not affect only adults, since it shows bimodal distribution among ages with a first peak of incidence in early childhood and a second peak in the elderly. Also, incidence varies in different geographical areas, with higher values in the areas with lowest socio-demographical index (SDI) [2]. Moreover, not only is sepsis a cause of high rates of in-hospital and early post-discharge deaths, but it also creates a burden in the survivors in terms of mobility, self-care autonomy, usual activities performance, persistence of pain and discomfort, and onset of anxiety and depression, leading to reduced quality of life and high use of healthcare resources, even after sepsis recovery [3].

During an infection, the pathogen triggers multiple and counterbalanced biochemical, hormonal, and immune reactions in the host. In sepsis, these reactions are no longer balanced, thus leading to harm to the host [4,5,6]. This definition makes clear that sepsis is a syndrome, with a still-uncertain pathobiology, resulting in a constellation of symptoms and signs which simultaneously involve several organs and systems, even if distant from the primary source of suspected infection [1]. However, even after decades of studying and defining sepsis, the mechanisms underlying this syndrome remain widely unclear, and the scientific society has failed to identify an unique biomarker for sepsis. In fact, an ideal biomarker is a measurable indicator giving accurate and reproducible information of the clinical conditions of a patient. Biomarkers include many molecules like metabolites, cytokines, chemokines, proteins, nucleic acids, and many others [7]. Besides the lack of a biomarker for sepsis, at present, available therapies—mostly consisting of support therapy for vital functions and antibiotics—act on a very limited part of the complex network of sepsis, with the patient’s organism being mainly responsible for the eradication of the infection. The high human and economic costs of sepsis raise the need to find complementary or alternative diagnostic and therapeutic strategies for two main reasons. The first one is antibiotic resistance. Drug resistant microbes are contributing to the rise of infection-related deaths [8,9], authorizations of new antibiotics are failing, and there is rising worry about the advent of bioterrorism agents [10] that may be difficult to treat with conventional therapies. The second one is the awareness that sepsis is a time-dependent syndrome whose optimal management consists of a prompt diagnosis, early and systematic application of evidence-based standard of care, and rapid address to higher level of care when appropriate [11]. This implies the necessity of managing and influencing the early phases of sepsis-related events. The purpose of this review is to analyze the molecular mechanisms of the earliest phases of sepsis to identify potential biomarkers for diagnosis, risk stratification, prognosis definition, general management, and therapeutic targets. Over recent years, there has been a great advancement in scientific knowledge in the field of early pathogenesis of sepsis, suggesting that we could be near to a revolution in our diagnostic and therapeutic approach to sepsis, a prospect for which every physician should be prepared. In this review, our attention has been addressed to the role of some potential biomarkers and therapeutic targets, namely the damage-associated molecular patterns (DAMPs) and pathogen-associated molecular patterns (PAMPs). DAMPs are endogenous molecules, while PAMPs are exogenous microbial products; both can trigger and enhance inflammatory responses through the activation of signaling pathways related to the pattern recognition receptors (PRR) [12], or other non-PRRs receptors.

## 2. Relevant Sections

Sepsis is enormously intensified by endogenous factors, including early involvement of both pro- and anti-inflammatory pathways. This model is made even more complicated by the peculiar characteristics of different pathogens and by host features like age, sex, genetics, comorbidities, chronic medications, environment, and concomitant lesions (i.e., trauma, previous surgery) [1].

The traditional sepsis model is biphasic, depicting an early hyperinflammatory cytokine storm and a later immune paralysis. However, the clinical evidence does not support sharp defined phases, portraying a more dynamic model where the two aspects of hyperinflammation and immune paralysis are not temporally consequential but are the moment-by-moment net result of coexisting, persistent, and competing pro- and anti-inflammatory pathways. This net result manifests with an early clinical picture characterized by fever, hypermetabolism, and shock, and a late one characterized by failure of primary pathogen clearance and occurrence of secondary infections [13,14].

### 2.1. DAMPs/PAMPs-Related Molecular Pathways in Sepsis

Every microorganism, during its life cycle, can release its constituents designated as microbe-associated molecular patterns (MAMPs) or pathogen-associated molecular patterns (PAMPs) in the case of virulent organisms. These PAMPs include essential parts of the pathogen like carbohydrates such as lipopolysaccharide (LPS) and mannoses; nucleic acids including DNA or RNA; peptides comprising microtubules and flagellin; wall molecules embracing peptidoglycan and lipoteichoic acid (LTA) [15].

PAMPs are recognized by two categories of immune receptors: the pattern-recognition receptors (PRRs) and non-PRRs.

Besides exogenous molecules, the immune system is also set to recognize endogenous molecules derived from a damaged host cell. These molecules—the DAMPs, also known as “alarmins”—are normally confined to the intracellular compartment, where they contribute to many homeostatic processes, and are released through various kinds of cellular damage like physical insults (such as the traumas or exposure to radiations), chemical agents (including various toxins or osmolality variations), metabolic changes (such as those related to ischemia and reperfusion), or exposure to infectious agents (like virus, bacteria, and protozoa) [13,16,17].

DAMPs include the high mobility group box 1 (HMGB1), the extracellular cold-inducible RNA-binding protein (eCIRP), the adenosine triphosphate (ATP), enzymes like the nicotinamide adenine dinucleotide (NAD), proteins and cellular molecules related to nucleic acids, such as heat shock proteins (HSPs), histones, members of the S100 family, cell free DNA (cfDNA), and mitochondrial DNA (mtDNA), recently grouped into the chromatin-associated molecular patterns (CAMPs) [13,18]. The DAMPs are recognized by many immune receptors that recognize PAMPs [13].

#### 2.1.1. Overview of the DAMPs

Expert authors in the field have defined the DAMPs as a “double-edged sword in sepsis” because of their ambivalent pro-inflammatory and immunosuppressive role, which might depend on their concentration, length of the exposure, and interaction with different receptors. Inside the cell, DAMPs are classified by their localization in the cytosol, nucleus, or mitochondria of endoplasmic reticulum (ER) [13]. Cellular harm can cause the release of DAMPs in the extracellular space through passive or active mechanisms. The first consist of various forms of cell death. Necrosis is characterized by cytosolic swelling and plasma membrane rupture and is the primary cause of passive DAMPs release. Apoptosis shows cytosolic shrinkage and preservation of plasma membrane integrity. For these features, it has been long considered a non-immunogenic form of cell death, but recent evidence suggests that it can be immunogenic under stress conditions, with subsequent DAMPs release. Also, the formation of membrane pores during pyroptosis has been related to DAMPs release in sepsis. Besides these three kinds of cell death, whose role in DAMPs release during is well defined, the contribution of other forms of cell death like necroptosis, ferroptosis, and NETosis is currently an object of study. Beside these passive mechanisms, some DAMPs have been reported to be actively released via lysosomal exocytosis and exosomes [19,20,21]. The main features of DAMPs, like their location and function in normal conditions, their prevalent mechanisms of release, their target receptors, and their role in inflammation, are synthetized in Table 1.

HMGB1—HMGB1 can be considered the prototypical alarmin [22,23,24,25]. It consists of 215 amino acids subdivided in three domains: the positively charged box A and box B, and a negatively charged acidic tail. For HMGB1, “location is the key to function”. In its usual location in the nucleus, it is involved in gene transcription. Post-translational modifications—including acetylation, phosphorylation, and methylation—of the Nuclear Localization Sequences (NLSs) of box A and B regulate the ability of HMGB1 to translocate to the cytoplasm during cellular stress, and then in the extracellular space [22]. When it is mobilized outside the nucleus, the protein acts as a modulator of inflammatory responses in many ways. First, it does so by interacting with both PRRs and non-PRRs [13,24]. Second, depending on its redox state, HMGB1 has a chemoattractant effect (reduced form), immunosuppressive effect (oxidizing form), or enhances the pro-inflammatory cytokines release (disulfide-bond possessing form) [22,23].

Histones—Histones are intranuclear proteins that contribute to the normal structure of chromatin and regulate gene expression. In humans, histones H2A, H2B, H3, H4, and DNA form the nucleosome complex. Histones can be released during necrosis or undergo post-translational modification during apoptosis. In the latter case, the modified histones leave the genomic DNA and translocate to the cytoplasm. Then, via still-unclear mechanisms, they are firstly exposed at the cell surface and then further released to the extracellular space. Alternatively, they can be actively secreted by exocytosis or become part of the NETs (neutrophil extracellular traps) [16]. Extracellular histones stimulate Toll-Like Receptors (TLRs), promoting pro-inflammatory cytokine synthesis and release [26]. On the contrary, dysregulation of the mechanisms of methylation, lactylation, and citrullination of extracellular histones has immunosuppressive effects via downregulation of dendritic cell-derived IL-12 release, M1 macrophage polarization to M2 phenotype, and protection from septic shock, respectively [13]. Moreover, histones and their fragments act as antimicrobial molecules with different pathogen species-specificity. Besides histones, we should remember that enzymes responsible of epigenetic regulation—i.e., histone methyltransferases (HMTs), histone demethylases (HDMs), histone acetyltransferases (HATs), histone deacetylases (HDACs), and peptide arginine deiminase (PAD)—are involved in many phases of the innate immune response [27].

CIRP—The Cold-Inducible RNA-Binding Protein (CIRP) is an RNA chaperone protein that, in its intracellular form (iCIRP), modulates the functions of messenger RNA. As suggested by its name, CIRP is induced by mild hypothermia but it is also passively released after several other insults such as hypoxic, oxidative, ischemia-reperfusion, and traumatic stresses [16,25]. Hypoxia causes the migration of CIRP from the nucleus to the cytoplasm. During stress conditions—including sepsis—iCIRP is released into the extracellular space becoming eCIRP. eCIRP has been suggested as a novel DAMP [13,25]. For example, eCIRP promotes the macrophage pyroptosis by interacting with a PRR, the nucleotide-binding domain-like receptor (NLR) family pyrin domain containing 3 (NLRP3). It also binds another PRR, the TLR4/MD2 (Toll-Like Receptor 4/Myeloid Differentiation factor 2) complex, which is pivotal in the activation of the NF-κB (Nuclear Factor Kappa B) signaling pathway for pro-inflammatory cytokine production. Moreover, eCIRP promotes the formation of NETs, an amalgam of nuclear chromatin, mitochondrial DNA, and neutrophil granule proteins, that mainly play a defensive role against lung infections. eCIRP also binds the non-PRR TREM-1, fueling inflammation with subsequent organ dysfunction and mortality in sepsis [13]. Besides pro-inflammatory eCIRP-dependent NF-κB signaling mechanisms, endotoxin tolerance and immunosuppressive effects have also been described. In particular, eCIRP has a strong affinity for the receptor of IL-6 (IL-6R). The interaction between eCIRP and IL-6R results in the expression of transcriptional repressors and corepressors inhibiting NF-κB gene reporters, and enhances the transcription of immunosuppressive genes for immune tolerance and suppression [25].

exRNAs—Extracellular RNAs (exRNAs) can be released by necrosis and apoptosis, but their inclusion within exosomes prevents their degradation by RNases and makes them more stable [16]. exRNAs can be double-strand (dsRNA) or single-stand (ssRNA) and bind to different receptors. dsRNAs bind the TLR3 and RIG-1 (retinoic acid-inducible gene 1) receptors, while ssRNAs bind the TLR7/8 [28].

cfDNA—Nuclear DNA is released in the extracellular space via various mechanisms. In sepsis, NETosis seems to be the prevalent one [16,29]. cfDNA triggers various signaling patterns through the interaction with several receptors, like TLRs (mainly TLR9), ALRs (Absent in melanoma-2-receptors), and RAGE (Receptor for Advanced Glycation End products) [30].

ATP—In normal conditions, adenosine triphosphate (ATP) is an intracellular source of energy and is present in the extracellular milieu (eATP) in very low concentrations. ATP can be released passively, but the two main mechanisms of release are through exocytosis and channel pores [16]. Once released, eATP can promote inflammation through the stimulation of the large family of purinergic receptors. The most studied purinergic receptors for eATP are the P2X and P2Y. P2 × 7 receptor, a subtype of P2X, is abundant in different immune cells, where it mediates the assembly of NLRP3 inflammosomes [13,31]. Once again, while high levels of this DAMP have been related to pro-inflammatory immune response, micromolar concentrations of eATP seem to have an immunosuppressive effect through the inhibition of LPS-stimulated IL-12, and macrophagic TNF-α secretion. Moreover, eATP-purinergic signaling stops when ATP is split by ectonucleotidases in adenosine, which promotes a negative feedback mechanism that confines local and systemic inflammation. Adenosine stimulates the A2 receptors to activate the canonical pathway of cAMP/PKA (adenylate cyclase-protein kinase A), which inhibits the NF-κB. The block of the NF-κB-mediated transcription inhibits the adhesion to endothelial cells (ECs) and causes the reduction of superoxide anion synthesis by neutrophils and the lowering of pro-inflammatory cytokines secretion [32].

NAD—The nicotinamide adenine dinucleotide (NAD) is traditionally linked to intracellular energy production. Many enzymes degrade extracellular NAD (eNAD) so that small amounts of it can be considered normal, but in stress conditions, the concentrations of this metabolite can increase. High levels of eNAD can engage purinergic P2X and P2Y receptors, triggering an inflammatory response like the one mentioned above for ATP. eNAD has enzymatic and non-enzymatic functions, described in many acute and chronic disorders, including obesity, cancer, and sepsis. NAD is actively released during exocytosis and diffusion through transmembrane transporters, or passively from necrotic cells. It is important to note that the many enzymes that can metabolize eNAD produce several metabolites that are involved in the further modulation of immune responses like immune cells activation, inflammatory genes transcription, and inflammosome assembly [33]. Moreover, nicotinamide phosphoribosyltransferase (NAMPT) and nicotinate phosphoribosyltransferase (NAPRT) have been proposed as DAMPs. These two intracellular enzymes catalyze the synthesis of NAD from its main precursors—nicotinic acid and nicotinamide—in their extracellular forms (eNAPRT and eNAMPT); they can bind to TLR4, triggering the differentiation and polarization of myeloid cells, the activation of inflammasome, and the secretion of cytokines [34,35].

HSPs—Heat shock proteins (HSPs) are molecular chaperones. HSPs are classified as high molecular and low molecular weight, expressed in KDa [36]. They can be passively released, actively secreted via ATP-binding cassette (ABC) lysosomes or granules, or mainly released by exosomes and ectosomes [16]. Compared to other alarmins, the knowledge on HSPs’ role in sepsis is limited, but there is evidence that they are involved in the initial stages of the disease. eHSP-60 and eHSP-70 seem to exert pro-inflammatory effects, whereas eHSP-27 has predominant anti-inflammatory effects. eHSP-27 modulates the expression of the IκB (inhibitor of nuclear factor kappa B) with subsequent reduction of inflammation, oxidative stress, and apoptosis; on the contrary, during the infection, eHSP-60 and eHSP-70 promote NF-κβ expression with subsequent increased release of IL-1β, TNFα, and other pro-inflammatory molecules [37]. The effects of HSPs seem to be mediated by the interaction with some TLRs [28], ad CLRs [38].

Mitochondrial-derived DAMPs (mtDAMPs)—Mitochondrial fragments like the mtDNA, the transcription factor A mitochondrial (TFAM), ATP, cardiolipin, cytochrome c, succinate, mtRNA, and mitochondrial N-formyl peptides (mtFPs) have been found to act as alarmins. As they preserve low-methylated CpG motifs, characteristic of microbial DNA, they are recognized not only by TLRs (in particular TLR9) but also by NLRP3 inflammasome, and are stimulatory of interferon genes (STING) [39]. mtFPs strongly promote the chemotaxis of immune cells and platelets that express formyl peptide receptors (FPRs) [40,41].

To make it even more complex, it is necessary to point out that the research on DAMPs is very enthusiastic and leads to the continued discovery of new information about these molecules and their function in health and disease [42,43,44]. Furthermore, different mechanisms may lead to different DAMPs’ release. For example, the loss of cell membrane integrity in necrosis can cause the release of mixed DAMPs, whilst apoptosis mainly leads to the release of nuclear DAMPs following DNA breakup. For these reasons the length of cell-free (cfDNA) released from apoptotic cells is ~180 bp, while cfDNA dispersed from necrotic cells can be as long as >10,000 bp. HMGB1 is in its hyperacetylated form if released by pyroptosis via inflammasome pathway, while it is not if it derives from necrotic or apoptotic cells. In addition, HMGB1 is in the disulfide form after pyroptosis, in the reduced or disulfide form after necrosis, and in the fully oxidized form (sulfonyl HMGB1) after apoptosis. Moreover, different stages of the same mechanism can release different DAMPs, as reported for apoptosis, where ATP is released at the pre-apoptotic stage, while HMGB1 is released at the late stage [16] (Table 1).

#### 2.1.2. Overview of the PAMPs

LPS—Lipopolysaccharide (LPS), also called endotoxin, is the main constituent of Gram-negative bacteria outer cell walls. However, the term LPS describes a class of molecules that share their chemical structure: the O-antigen, the 3-deoxy-D-manno-octulosonic acid (KDO) and heptose-containing core oligosaccharide, and the lipid A [45,46]. The lipid A portion is the “endotoxic principle” of LPS, and its endotoxic activity depends on its structure, length, number, and distribution of lipid chains, as well as on the phosphorylation status of each sugar unit [46]. Each bacterial species has its own group of LPSs, varying in their composition. LPS interacts with a variety of proteins and enzymes—such as the lipopolysaccharide-binding protein (LBP), various cluster differentiation (CD14, CD16, CD18), antibodies, hemoglobin, lysozymes, and lactoferrin [45,47]. However, extracellular LPS mainly activates the TLR4. Recently, it has been highlighted that LPS may enter the cell via a clathrin-mediated endocytosis of outer membrane vesicles (OMVs), or through the preliminary extracellular binding to the HMGB1 protein, and subsequent conjunction with RAGE [45].

Flagellin—Flagellin has been recently reported to be a Gram-negative PAMP that triggers the TLR5. A study by Liaudet et al. has highlighted that flagellin induces the expression of pro-inflammatory cytokines like IL-8 and of pro-adhesive molecules such as intercellular adhesion molecule-1 (ICAM-1) in vitro, and that intravenous administration of flagellin in mice causes a severe lung inflammation, with a stronger chemotaxis and lung hyper-permeability than the one caused by LPS. These effects have been related to the nuclear translocation of the transcription NF-κB in the lung. In patients with sepsis, plasmatic flagellin has a significant positive correlation with the extent of lung injury, the alveolar–arterial oxygen gradient, as well as with the duration of the sepsis [48]. On the contrary, other studies have suggested an anti-septic role of flagellin mediated of TLR3, TLR4, and IL-1RN, and upregulation of vascular cell adhesion protein (VCAM). These data suggest that a pretreatment with flagellin might facilitate endothelial repair and thereby promote survival following sepsis [49]. Moreover, flagellin seems to enhance the phagosome formation and increase reactive oxygen species (ROS) levels in macrophages, in a TLR5-dependent manner, with beneficial effects in abdominal-sepsis murine models [28,50].

#### 2.1.3. Overview of PRRs, Non-PRRs, and Related Pathways

PRRs are a vast group of receptors capable of recognizing PAMPs and are expressed constitutively in the host, mainly on immune cells, but also on somatic cells. They are encoded by a limited number of germ line genes, and typically non-clonally distributed. PRRs are divided into five sub-families: the Toll-like receptors (TLRs), the nucleotide-binding oligomerization domain (NOD)—Leucine Rich Repeats (LRR)—containing receptors (NLR), the retinoic acid-inducible gene 1 (RIG-1)-like receptors (RLR; also known as RIG-1-like helicases—RLH), the C-type lectin receptors (CLRs), and the Absent in melanoma-2-like receptors (ALRs) [51,52]. PRRs are expressed on membranes—both in the cell surface and in the intracellular compartment—and in the cytoplasm. Their structure is composed of three domains: ligand recognition, intermediate, and effector. After binding to the recognition domain, the ligand activates, through the effectors domain, signaling pathways that produce effects like recruiting and releasing of cytokines, chemokines, hormones, and growth factors; induction of acute or chronic inflammation; initiation of the innate immunity and subsequent acquired immune response; elimination of dead or mutated cells [7,52,53].

TLRs—To date, 11 TLRs have been described in humans, and some—if not all—of them are probably involved in the recognition of the major PAMPs and DAMPs. Some TLRs (TLR1, 2, 4, 5, 6, and 10) are expressed on the surface of immune cells, as hetero- or homodimers, and they are specialized in the recognition of bacterial products [52,54,55,56]. Table 2 and Figure 1 and Figure 2 report an overview of the features of TLRs, like their normal location, morphology, and main ligands and signaling mediators/adaptors. [52,56]. Despite the initial notion of strength association between one particular TLR and its microbial ligand, it is now accepted that these receptors might combine in a wide repertoire able to distinguish closely related ligands [28]. Also, there is preliminary evidence that polymorphisms in Toll family proteins might explain the variability in individual responses to similar infective triggers [54,57].

Interestingly, as described for other molecules involved in the pathogenesis of sepsis, the expression of TLRs and their related cytokines is a dynamic mechanism in the different phases of the illness [57]. In fact, beside their pro-inflammatory function, some authors have reported that TLRs are also implicated in the so-called “compensatory anti-inflammatory syndrome” (CARS), also known as “immunoparalysis” or “immune reprogramming”, in the early phase of sepsis [5,14,55,58]. Several molecular mechanisms negatively regulate TLR-induced signaling. These host-derived negative regulators of TLRs can act through three main mechanisms: detachment from the adaptor molecule complex, demolition of proteins involved in TLR signaling, and transcriptional regulation of the signaling pathway [55,58]. Additionally, there is evidence that, even if most TLRs are expressed ubiquitously, many of them show tissue-specific patterns of expression with high concentrations of TLRs-mRNA (Table 3). It is of no surprise that the highest concentrations of TLRs-related mRNA have been mostly detected in tissues that have contact with the external environment or a crucial role in immunity [59].

NLRs—As some pathogens cause infection of the cytoplasm, some PRRs have to be intracellular. These PRRs are the NLRs [52], and their structure is illustrated in Figure 3A. The presence of a PYD or CARD domain allows the subdivision of the family into NLRP or NLRC receptors, respectively. Every NLR—NLRP1, NLRP3, NLRP6, NLRP7, NLRP12, and NLRC4—forms its own inflammasome [60]. Inflammosomes are cytosolic multiprotein oligomers that mediate the mechanism of pyroptosis, which is implicated both in the inflammatory response to pathogens and the damage to the host. The most characterized one is the NLRP3 inflammosome (or “canonical inflammosome”). The assembly of the canonical inflammasome requires two steps, priming and activation, mediated by various stimuli (Figure 3B). The result of these two steps is the activation of the final effector, the caspase-1, which cleaves the pro-interleukins (pro-IL-1 and pro-IL-18) and the gasdermin (GDSMD) in their mature forms. IL-1β and IL-18 recruit macrophages and neutrophils in the site of infection, whilst mature GDSMD forms membrane pores that cause lytic cell death [31,61]. The non-canonical pattern of pyroptosis is mainly activated by LPS and involves caspases-4/5/11 which also finally cleave GSDMD [62,63]. On one hand, pyroptosis has positive effects on the clearance of infected cells; on the other hand, it contributes to the hyperinflammation in sepsis [12] and to the pathogenesis of many chronic inflammatory diseases [31,60].

Pyroptosis is not an isolated mechanism since other pathways (such as autophagy) regulate it. Autophagy is a pivotal homeostasis mechanism that consists of the selective degradation of macromolecules and damaged organelles via the action of lysosomes. It has been reported that autophagy can downregulate pyroptosis through the clearance of DAMPs and PAMPs. In fact, mouse models of sepsis, lacking essential genes of the autophagy process—i.e., Atg7, Atg8, and Atg16L1—show worse inflammatory injuries and shortened survival because of the reduced pathogen clearance resulting from the high activation of both canonical and non-canonical pyroptosis pathways with elevated levels of IL-1β and IL-8, high release of inflammatory cytokines, and enhanced activity of inflammasomes [12].

The described mechanism seems linear and logical and designs pyroptosis as a part of innate immunity. However, recent studies have demonstrated that the network of immunity is more intricate than it seems. For example, not all types of inflammasomes are involved in the recruitment of phagocytes. NLRP3 does not recruit phagocytes, as demonstrated in NLRP3-deleted mice undergoing polymicrobial sepsis after a CLP. Moreover, IL-1 can be secreted by neutrophils and macrophages in a non-pyroptotic process. Besides this evidence, pyroptosis has also been studied for its role in adaptive immunity. In fact, IL-1 and IL-18 are well-known stimulators of natural killer, T, and B lymphocytes. Furthermore, these cells may undergo pyroptosis themselves, with subsequent immunosuppression and release of DAMPs that triggers an inflammatory vicious circle [64,65].

RLRs—The RIG-I-like receptors (RLRs) family includes three innate immune receptors: the retinoic acid inducible gene I (RIG-I), the melanoma differentiation-associated protein 5 (MDA5), and the laboratory of genetics and physiology 2 (LGP2). They are all RNA helicases, but while RIG-I and MDA5 contain a CARD domain, LGP2 is a CARD-free structure. For this reason, it does not activate a transduction pathway, but it is a regulator of the RLR signaling pathway. These receptors recognize viral RNA and initiate the antiviral responses [66] (Figure 4). Besides PAMPs, some DAMPs, as self-RNAs, can also activate RLRs. Even if the role of these receptors in anti-viral response—and in some kind of cancers—has been widely described, their implication in sepsis is less clear [67].

CLRs—C-type lectin-like receptors are transmembrane receptors, primarily found on myeloid cells. They recognize both PAMPs and DAMPs and can activate or inhibit receptors. In fact, their intracellular tail can either present a classical immunoreceptor tyrosine-based activating motif (ITAM) in the intracellular tail, interact with ITAM-containing adaptor proteins, or contain a hemi-ITAM motif (Figure 5). Surprisingly, the same CLR can integrate distinct positive and negative signals according to the ligand or the environment characteristics [68] (Figure 5). The role of these receptors in infection has been widely reported. Regarding sepsis, some of them have been described as mediators of the sepsis-induced coagulopathy. C-type lectin-like receptor 2 (CLEC-2) is expressed on platelet membranes, and its soluble form (sCLEC-2) has been receiving attention as a predictive marker for thrombosis [69].

ALRs—They have been recently identified as PRRs involved in cytosolic and nuclear pathogen DNA recognition [70]. The interaction with the ligand causes the formation of a functional inflammasome. The result is an upregulation of IL-1β and IL-18 production and increase in the levels of interferon. In recent years, members of the ALR family have been related to autoimmunity [71], but their role in sepsis has not been clarified yet.

Non-PRRs are also involved in the reaction to PAMPs and/or DAMPs, but do not share the common features of PRRs; they include receptors for advanced glycation end products (RAGE), triggering receptors expressed on myeloid cells (TREM), and G-protein-coupled receptors (GPCRs).

RAGE—The RAGE are multiligand transmembrane receptors, members of the immunoglobulin superfamily, expressed in many tissues, involved not only in the pathogenesis of inflammation and infection but also that of diabetic complications, atherosclerosis, carcinogenesis, and neurodegenerative diseases. The receptor consists of three extracellular domains (V, C1, and C2), a transmembrane domain, and a short cytosolic tail. Among RAGE ligands, we find exclusively DAMPs such as advanced glycation end products (AGEs), HMGB-1, amyloid-b peptide, and others [72]. They can bind one or more sites of the V domain, which is structurally like the variable domain of the Fab fragment of human immunoglobulin. The different modality of the ligand-extracellular domain may account for the activation of different pathways such as NF-κB, mitogen-activated protein kinase, and other kinase-mediated responses [73]. Interestingly, even if it has been reported that RAGE only binds to DAMPs, recent studies have revealed that there is a crosstalk involving RAGE and TLRs, and that three canonical RAGE ligands—AGEs, HMGB1, and S100—activate both TLRs and RAGE [72].

TREM—TREM receptors include TREM-1, TREM-2, TREM-3, and TREM-like transcript-1 and 2 (TLT-1, TLT-2). They show a variegated response to LPS. TREM-1 is a member of the immunoglobulin superfamily and is upregulated by LPS. TREM-2 is downregulated by LPS and attenuates the inflammatory response. TREM-3 has been found only in mice [74].

GPCRs—Among G-protein coupled receptors, we include the formyl peptide receptors (FPRs). In humans, three FPRs have been identified, namely FPR1, FPR2/ALX, and FPR3, expressed in a variety of cells, with the highest expression in neutrophils for FPR1 and FPR2, and monocytes/macrophages for FPR3 [39].

#### 2.1.4. Effects of DAMPs and PAMPs Release during Sepsis: Cellular, Tissue, and Organ Level

Immune cells—Many studies have analyzed the PAMPs/DAMPs and receptor expression in the immune cells, mainly in macrophages and neutrophils. For example, iHMGB1 seems to prevent macrophage cell death in bacterial infection by mediating autophagy, whilst eHMGB1 induces cytokine release via the TLR4/MD2/MyD88/NF-κB pathway [75,76]. Moreover, when activating the RAGE pathway, it induces inflammasome and caspase-1 activation with subsequent pyroptosis [22]. The exposure of macrophages to the LPS has been shown to cause the translocation of the cytosolic enzyme ACLY, with subsequent histone modification and activation of the NF-κB pathway [77]. Thanks to the studies on sepsis, the assumption that neutrophils are a short-lived, terminally differentiated homogeneous population has been questioned. In fact, recent studies have revealed that besides the N1 and N2 subsets of neutrophils, several new subsets including aged, antigen-presenting, and reverse-migrated neutrophils have been described, which potentially contribute to the pathogenesis of sepsis based on their pro-inflammatory and immunosuppressive functions. This heterogeneity seem to depend also on DAMPs [22,78]. For example, HMGB1 can activate both TLR4 and RAGE pathways in neutrophils; the first activates the NADPH oxidase activity, which is essential for bacterial killing, whilst the second reduces the NADPH oxidase function. However, HMGB1/RAGE-mediated neutrophil NADPH oxidase dysfunction has been reported to be connected to both higher survival of septic shock and reduced bacterial clearance [79]. These conflicting results suggest that balance of these interactions may be crucial in determining defensive or harmful effects in sepsis, and may in part be determined by the redox state of HMGB1 [22,80].

Non-immune cells—As already stated, sepsis is a syndrome simultaneously involving several organs and systems. This assumption is well summarized in one of the most used scores for sepsis prognosis, the SOFA (Sequential Organ Failure Assessment) score. In this score, the criteria for the assessment of sepsis explore clinical, laboratoristic, and therapeutic features belonging to the respiratory, nervous, cardiovascular, hepatobiliary, coagulation, and urinary systems [81,82]. It is of no surprise that many studies have reported that DAMPs/PAMPs pathways are able to disrupt the homeostasis of all those systems (Table 4).

### 2.2. Actual and Potential Applications of DAMPs/PAMPs and Related Pathways in Clinical Practice

#### 2.2.1. Biomarkers for Diagnosis, Severity Stratification, and Prognosis of Sepsis

To explore the possible role of DAMPs and PAMPs as biomarkers for sepsis, we should reiterate the characteristics of a biomarker as a measurable, accurate, and reproducible indicator of normal and pathogenic biological processes, with both high specificity and sensitivity.

Over the years, the clinical markers for sepsis have been changing and becoming more accurate. Traditionally, leukocytosis and high concentrations of C-reactive protein (CRP) have been related to infection and, together with clinical features, have been used to identify sepsis and drive therapeutic decision-making. However, it is known that the WBC (white blood cell count) and the concentration of CPR can be affected by many factors such as steroid therapy, surgery, smoking, trauma, and others. Recent studies have been questioning the role of WBC and CRP in sepsis. For example, it has been highlighted that in urosepsis, a WBC higher than 14 × 10^9^/L can be used as an early warning factor, but also that WBC can be normal or reduced, together with platelets and fibrinogen, because of a process of sepsis-induced consumption [110]. In 2016, Pradhan et al. evaluated the CRP value of 64 patients fulfilling the criteria for SIRS (systemic inflammatory response syndrome) admitted to ICU (with 51 of them further diagnosed with sepsis), and reported, for this marker, a sensitivity of 84.3%, a specificity of 46.15%, positive predictive value (PPV) of 84%, and negative predictive value (NPV) of 42.8%, with the best diagnostic accuracy at 61 mg/L [111]. Another marker is IL-6, a cytokine synthesized by T lymphocytes, fibroblasts, endothelial cells, and monocytes. Besides its functions in immune regulation, hematopoiesis, and oncogenesis, it is an acute phase cytokine involved in inflammation and sepsis. Its normal serum concentration is <5 pg/mL, and it rapidly increases after infection, with a peak within 2 h [112] that often precedes the fever and the rise in CRP and PCT, and reaches values > 500 pg/mL in sepsis [110]. Also, values higher than 80% have been reported for both high sensitivity and specificity in the diagnosis of sepsis [113]. Procalcitonin (PCT), the propeptide of calcitonin, emerged as a biomarker for sepsis because of its sensitivity and specificity, reported to be higher than those of CRP [114]. In normal conditions, serum PCT concentration is <0.1 ng/mL, but in septic patients, the cells of many parenchymal organs (liver, kidneys, adipose and muscle tissue) produce large amounts of PCT. Its increase starts in the first hours after the bacterial infection; it becomes detectable after 4 h, peaks at 6 h, and reaches a plateau at 8–24 h [115]. In the years, authors have been studying the variation of sensitivity and specificity of PCT in the context of different types of sepsis (e.g., CAP-associated, urosepsis) and have explored the cut-offs for clinical decisions, mostly regarding the decision to start, maintain, and dismiss antibiotic therapy [116]. Among its limitations, mild elevation in the case of hepatic or renal impairment and after major surgery have been reported [110]. Under stress conditions like hypoxia and hypoperfusion, cells switch from aerobic metabolism to anaerobic, leading to the production of lactate (Lac). The normal value of serum Lac should be <2.0 mmol/L. Even if there is no Lac value that could be considered diagnostic for sepsis—since Lac can also rise in other situations like cell lysis in some tumors or trauma—in septic patients, Lac concentrations higher than normal have been related to poor prognosis, while its clearance in the first hours after fluid resuscitation has been associated with higher 28-day survival rates [117]. Lately, other markers have caught attention and are emerging in clinical practice, like proadrenomedullin and presepsin [118]. The mid-regional fragment of pro-adrenomedullin (MR-proADM) derives from the degradation of adrenomedullin (ADM), a peptide produced by all cells, but mostly by adrenal medullae, cardiac atria, and lungs. ADM is essential for the function of the endothelial barrier and the vascular tone [118,119]. Li et al. performed a meta-analysis including 11 studies for a total of 2038 cases of sepsis. They reported a sensitivity of 0.83 (95% CI: 0.79–0.87) and a specificity of 0.90 (95% CI: 0.83–0.94), an area under the curve (AUC) of 0.91, and the best cut-off value for MR-proADM diagnosis of sepsis at 1–1.5 nmol/L [120]. Presepsin is the N-terminal fragments of the soluble form of a CD14 subtype (sCD14-ST). CD-14 is involved in LPS interaction with TLR4. The values of presepsin differ in healthy controls (294.2 ± 121.4 pg mL^−1^) when compared to septic patients (817.9 ± 572.7 pg mL^−1^) [121,122]. In a meta-analysis by Han et al., including 11 publications with 3106 patients, the pooled sensitivity was 0.83 (95% CI: 0.77–0.88), specificity was 0.81 (95% CI: 0.74–0.87), positive likelihood ratio was 4.43 (95% CI: 3.05–6.43), and negative likelihood ratio was 0.21 (95% CI: 0.14–0.30). The area under the curve (AUC) was 0.89 (95% CI: 0.86–0.92) [123]. Presepsin and MR-proADM have shown good values of accuracy in another meta-analysis by Liang et al. [124].

Our rapid review on the existing, most-used markers for the diagnosis of sepsis aims at showing what qualities markers should have to be useful for clinical practice: a cut-off value, a predictable trend during the progression of the pathology (determining an ideal timing for measurement and monitoring), and known sensitivity and specificity. Unfortunately, we have not been able to find—to date—any study which allows us to consider any of the mentioned DAMPs or PAMPs as a suitable biomarker for the clinical management of sepsis in humans. In fact, even if there is evidence of a relation between their increase—or decrease—and the likelihood of a diagnosis of sepsis, we have not found any precise cut-off, any recommended measurement technique, nor any indication about the optimal biologic sample, the ideal timing for the first assay or for serial dosing, or data about their sensitivity and specificity.

However, as regards the risk stratification and prognostic significance of DAMPs, there is interesting evidence of their potential applications. Histones and HMGB1 protein levels have been proposed as indicators of prognosis, but studies have provided divergent results [84,87]. Sawada et al. investigated the relationship between serum histone H3 and HMGB1 levels, the illness severity score, and the prognosis in postoperative patients in the ICU. They found that these two DAMPs positively correlated with the SOFA score [125]. Alcamo et al. proposed that the measurement of HMGB1 in the plasma of septic pediatric patients might be a predictor of MOF, with mild sensitivity (55.3%) but good specificity (90%) [126]. In community-acquired pneumonia (CAP), some authors found that high blood levels of HMGB1 were correlated with the severity of the disease, showing higher levels in higher pneumonia severity index (PSI) risk classes, while other authors reported similar levels of this alarmin in CAP patients with or without sepsis. To explain these divergences, considering that there is evidence that the release of HMGB1 predominantly occurs at the site of infection, Alpkvist et al. remarked that HMGB1 levels measured in lower respiratory secretions might better correlate with the severity of the disease, and that the specific pathogen might also affect the levels of this DAMP. *S. pneumoniae* being the most common etiology of CAP in all severity classes, they analyzed local (sputum) and systemic (plasma) HMGB1 concentrations and found that the levels were significantly higher in patients infected by *S. pneumoniae* compared to other etiologies, while no correlation was found between HMGB1 levels in plasma and sputum and between HMGB1 concentration and pneumonia severity [127]. eCIRP levels seem to positively correlate with sepsis severity and mortality. Interestingly, in a study on 69 adult patients with sepsis, eCIRP levels significantly were correlated with the Acute Physiology and Chronic Health Evaluation II (APACHE II) score, the SOFA score, the serum creatinine level, and the procalcitonin level. When the eCIRP concentration in the peripheral blood was greater than 10 ng/mL, the mortality risk increased by 1.05-fold [128]. Among NAD-related enzymes, there is little evidence of the role of extracellular eNAPRT as an emerging biomarker of sepsis and septic shock [33,34]. Experiments on animal models of sepsis, induced by *E. coli* injection CLP, have highlighted increased concentrations of cell-free DNA (cfDNA) [129,130]. Similar observations have been reported in human patients with sepsis, where cfDNA levels not only rise in the very first hours of sepsis but are also correlated with higher SOFA score—and subsequently disease severity and mortality. Moreover, several fragments of cfDNA with different length were identified, probably correlating with different sources of the alarmin (i.e., several organs and tissues) but also with the severity of the hypotensive stress [131]. These studies on ICU patients have highlighted the high sensitivity and specificity of this DAMP for the rapid identification of high-risk patients [131,132]. A study including 31 preterm neonates with suspected sepsis showed that cfDNA, DNase I, nucleosome, and CRP concentrations were higher than those measured in non-septic preterms [133]. Also, higher plasma levels of S100A8/S100A9 and S100A12 measured in the first 24 h from ICU admission in 49 septic shock patients with similar SOFA scores were correlated with higher mortality at 28 days [134]. Circulating cell-free mtDNA, measured via quantitative PCR (qPCR), has been associated with the overall 28-day mortality in critical patients [135], including those with sepsis. In a study from Yamanouchi et al., concentrations of mtDNA peaked on the day of admission (day 1) in patients with trauma, whereas they increased on day 1 and remained constant until day 5 in septic patients. Additionally, mtDNA levels on day 1 were significantly higher in non-survivors compared with survivors of trauma (*p* < 0.05), but not of sepsis [136]. Wang et al. reported that in 107 patients hospitalized in the emergency department (ED) for sepsis (n = 72) or septic shock (n = 35), the median mtDNA level was significantly higher in the septic shock patients, correlated with the lactate concentration and SOFA score but not with CRP and PCT levels. Moreover, mtDNA and lactate levels in non-survivors were significantly higher, with the mtDNA levels having a superior prognostic prediction value than that of lactate levels [137]. Other studies reported similar findings [138,139]. In contrast to mtDNA, only a few publications can be found on the role of mtFPs in sepsis. In these studies, mtFPs have been related to vascular leakage, cardiovascular collapse, hyperthermia, blood clotting, and lung injury [140,141]. A study from Canul-Euan et al. on early onset sepsis in newborns revealed that the concentration of eHSP-27 (27 kDa) was 2.5-fold lower in septic babies compared to the healthy ones, while eHSP-60 and eHSP-70 increased 1.8- and 1.9-fold in the group of septic babies [142]. Other authors reported an increase in eHSP-72 eHSP-90 during sepsis [143,144,145]. In a chronic kidney disease (CKD) murine model, in which sepsis was induced, more severe ALI was related to a lower expression of HSP-70, suggesting that this alarmin could be a predictor of SA-ALI [146]. Similarly, other studies have suggested that HSP-70 induced attenuation of lung injury in sepsis [147,148,149]. This anti-inflammatory effect of HSP-70 has been related to stabilization of IκB through preventing IKK (IκB kinase) activation in respiratory epithelium [150,151].

Beside DAMPs/PAMPs detection, other authors have focused on receptor detection and clinical significance. TLR4 hyperexpression has been identified as an indicator of severe prognosis during SARS-CoV-2 infection [152]. Lenz et al. reported evidence for prognostic properties of neutrophil TLR2 and TLR9 expression in predicting 30-day mortality in unselected critically ill patients, [79]. Elevated levels of serum soluble RAGE (sRAGE) correlate with sepsis severity [73]. sTREM-1 levels correlate with the severity of organ dysfunction, assessed with the sepsis-related organ failure assessment (SOFA) score [82], and with 28-day mortality [153].

#### 2.2.2. Treatment Strategies

Even after great improvements in the management of sepsis, the mortality from septic shock remains high, leading to great focus on research into alternative or complementary treatments. After our review of the literature, we find it useful to classify these strategies as unselective and selective. The first ones find their presumptive effectiveness in a general ability to remove PAMPs and DAMPs from the bloodstream; the second ones are characterized by targeted inhibition or stimulation of PAMPs/DAMPs related pathways.

The first group of approaches refers to the extracorporeal blood purification (EBP) therapies [154]. It is important to note that it is difficult to compare data from different studies because of the lack of large-scale randomized clinical trials and great variability in clinical practice all over the world. In the effort to make uniform the use of EBP techniques, the terminology has recently been the object of revision [155,156,157,158]. The EBPs use an extracorporeal circuit to remove and/or regulate the concentration of circulating substances, and eventually support the functionality of specific organs. The different types of EBPs are classified depending on the mechanism of solute removal—including diffusion, convection, adsorption, and centrifugation, or various combination of them. The major groups are the whole blood therapies (e.g., CRRT, continuous renal replacement therapy; ECMO, extracorporeal membrane oxygenation), the plasma therapies, subdivided into plasma exchange (e.g., plasmapheresis) and plasma absorption (e.g., PAF, plasma adsorption filtration; PFAD, plasma filtration adsorption dialysis), and the albumin-based therapies. CRRT is based on the mechanisms of convection and/or diffusion for indirect solute removal. The techniques of hemoadsorption (formerly referred to as hemoperfusion) are based on the passage of blood through a sorbent-containing cartridge or a hemofilter, responsible for selective or broad-spectrum solute removal. Some of the targets of hemoadsorption include endogenous toxic molecules (e.g., bilirubin, myoglobin), endotoxins, and inflammatory cytokines [157,159].

Regarding the role of EBPs in septic patients, different techniques are known to be of some utility thanks to their ability to remove endotoxins and cytokines [160], but also have several limitations due to many issues including complications (e.g., coagulopathy, risk of severe electrolyte disorders), need for expertise, and high costs [161,162]. Lately, some authors have started exploring their potential capacity to remove lethal levels of DAMPs and PAMPs [84,163]. The surface-treated polyacrylonitrile-co-methallyl sulfonate membrane (AN69ST), already known for its ability to remove IL-6 and lactate from the blood of septic patients, has demonstrated good capacity to decrease the concentration of HMGB1 when compared to polymethylmethacrylate (PMMA), polyethersulfone, and polysulfone membranes. Unfortunately, studies on AN69ST have only been conducted in vitro [164,165], whereas in vivo trials are lacking; for PMMA, some studies on animal models are available [166]. The polymyxin B cartridge (PMX) has demonstrated the ability to reduce the concentration of endotoxin in vitro, but its benefit for the survival of septic patients—investigated in three main trials—is controversial. The EUPHAS trial, conducted on 64 patients with Gram-negative abdominal septic shock, reported that—when compared to the conventional therapy—the use of PMX hemoadsorption increased the mean arterial pressure (MAP), reduced the requirement of vasopressors, and improved the SOFA score [167]. On the contrary, the ABDOMIX, including 243 septic shock patients with peritonitis, failed to demonstrate a reduction in the 28-day mortality in the PMX group compared to the control group [168]. The EUPHRATES trial, on 450 septic shock patients, selected after endotoxin activity assay (>0.60 or higher), initially showed similar results [169]. A post hoc analysis of the subgroup of patients with endotoxin activity of 0.60–0.90 revealed an improvement of the 28-day survival rate, suggesting that the efficacy of the treatment may be affected by the appropriate selection of patients [170]. Interestingly, retrospective studies have revealed that, besides its ability to adsorb the endotoxin, the PMX also removes HMGB1. Sakamoto et al. reported that, in septic shock patients (n = 20), the PMX treatment causes hemodynamic improvement and SOFA score reduction, but the study did not mention any data on overall mortality [171]. Other authors have explored the ability of PMX to reduce HMGB1 in sepsis but, again, the effects on the outcome have not been investigated [84]. Another hemoadsorption cartridge, CytoSorb^®^, has been proposed for the treatment of critical patients—including septic patients—for its ability to remove endotoxin, bilirubin, myoglobin, and various cytokines [172]. Recent studies have highlighted that this cartridge also adsorbs DAMPs like HMGB1, histones, and S-100, but in this case, its role in improving the outcome of sepsis is also controversial. Gruda et al. evaluated the performance of this cartridge in vitro, reporting that it is able to significantly adsorb cytokines (i.e., MIP1-α, IL-6, and IFN-γ), PAMPs (α-toxin, SpeB, and TSST-1), and DAMPs (C5a, HMGB-1, and S100-A8) [161]. Chen et al. tested a baboon model of pneumococcal pneumonia and sepsis with organ dysfunction, treated with 24 h administration of ceftriaxone, and then randomized to blood purification using a filter coated with heparin sulfate (n = 6) or sham treatment (n = 6) from 4 to 30 h after inoculation. Among the results, they noticed a decrease in the concentration of peripheral blood pneumococcal DNA, and a lower activity of the NLPR3 inflammasome. The study is of interest for the choice of a nonhuman primate model, since baboons have larger size allowing invasive monitoring, resemble humans in their lung anatomy and posture, and show similar hemodynamic and immunologic sepsis pathways. However, since the animals were euthanized at 48 h after the inoculation—to perform the necropsy and collect tissue samples—there are no data regarding the impact of the treatment on the mortality. Interestingly, the authors report a peaking of *S. pneumonia* DNA concentration 12 h after the administration of ceftriaxone. Since beta-lactams are bactericidal, the author wonder if the use of bacteriostatic agents in the first phase of sepsis may lower the release of PAMPs—including the endotoxin, which is recognized to be one of the major mediators of cardio-circulatory collapse in sepsis [173]. In a case series of post-neurosurgery septic shock patients, the combination of CRRT and CytoSorb caused permanent or transient clinical improvement, but even if the authors hypothesize a role for DAMPs and PAMPs removal, none of them has been measured [174]. Even if CytoSorb has been approved for the treatment of critical patients since 2011, some meta-analyses have found no significant impact on mortality, so the authors conclude that there is not striking evidence to recommend its systematic use outside clinical trials [175,176,177].

The evidence suggesting that the removal of a single septic mediator is ineffective in reducing the mortality, and that available hemoadsorption cartridges show unselective adsorption activity, has induced some authors to develop and investigate the therapeutic potential of telodendrimer nanotraps (TD-NTs). Telodendrimers are architectural polymers consisting of a linear polymer and a hyperbranched structure (dendron). The extremities of the dendron bind to positively or negatively charged molecules, which act as nanotraps (NT+ and NT−, respectively). The assumption is that, since pro-inflammatory molecules (e.g., TNF-α, IL-1, IL-6, and HMGB1) are negatively charged and anti-inflammatory molecules (e.g., IL-11) are positively charged, the use of customized nanotraps could selectively remove these inflammatory mediators. Shi et al. firstly studied the effect of intra-abdominal TD-NT administration in a murine septic model of CLP, with different timing. Given that the mortality rate 48 h after CLP was 62.5%, the administration of NT+ immediately after CLP surprisingly increased the mortality rate (77.8%), hypothetically because the early removal of pro-inflammatory molecules would inhibit the innate immune response. On the contrary, the concomitant administration at time 0 of NT+ and NT− slightly improved the survival rate (37.5%), whilst their application at 3 h and 8 h significantly improved the survival rate (50–62.5%). Then, they combined the use of an association of NT+/NT− with the antibiotic therapy (imipenem/cilastatin 50/50 mg/kg, 50% of its full dose in mice, to mimic septic hypoperfusion) administered 3 h after CPL. The combination was able to reach a 100% survival rate before euthanasia on day 42. The study has some important limitations, mainly in the fact that nanotraps were directly administered in the abdominal cavity, which is not acceptable for clinical use, and that the effects of the treatment on the later phase of septic immunosuppression and the long terms effects of the TD-NTs on the immune system have not been investigated [178]. Another strategy is that of nanoparticles-based vaccines, like the ciVAX. This vaccine consists of paramagnetic beads covered with a broad-spectrum engineered opsonin (FcMBL, Fc-mannose binding lectin) that rapidly binds to PAMPs, like glycoproteins and glycolipids, that are found in more than 120 species of pathogens and toxins. The complete vaccine is obtained by mixing beads with mesoporous silica (MPS) rods covered with GM-CSF and CpG-rich oligonucleotides. In a study by Super et al., the vaccine was administered to mice subcutaneously, together with *E. coli* fragments. The MPS rods rapidly formed a GM-CSF/CpG-releasing matrix, attracting immature dendritic cells and activating them. The mature DCs then migrated to the draining lymph node and interacted with resident B and T cells to start an adaptive response to the bacterial antigens. The authors then investigated the effect of ciVAX prophylaxis in mice challenged with a lethal intraperitoneal administration of the homologous antibiotic-resistant *E. coli* RS218 strain (O18:H7). At day 35, 9% of the unvaccinated mice vs. 100% of the vaccinated ones survived. Moreover, the immune response seemed to be prolonged, as a single injection protected >90% of mice from a new challenge at day 90 after the vaccination. The prophylaxis was tested also in pigs, challenged with intravenous injection of a human isolate of *E. coli* 41,949 (OM:H26), and all of them (n = 4) survived to the end of the experiment (euthanasia at 28 h for n = 2, and 72 h for n = 2) [179].

With regard to the selective strategies, at the moment, the most studied consist of antibody neutralization, competitive antagonism, and enzymatic inactivation [180] (Table 5).

Clinical data have shown higher survival rates in septic shock patients who generated HMGB1 autoantibodies, and administration of HMGB1-neutralizing antibodies not only altered the early phase inflammatory response but also reduced the susceptibility to secondary infection [181]. Yang et al. reported that HMGB1 antagonists—like the recombinant HMGB1 A box or the P5779 peptide derived by the HMGB1 box B—are protective against sepsis mortality [182,183,184].

Heparin is one of the most used histone-neutralizing agents, with an effect of preventing histone-mediated coagulation, but it also alleviates HMGB1-induced inflammation. However, heparin is not routinely used in critically ill patients, as it unfortunately increases the risk of fatal bleeding in those with coagulopathy [84].

Histones and their fragments are promising future therapeutic strategies, mainly because of their species-specific antimicrobial activity; the main limit to their systematic use is that they are themselves cytotoxic, and their injection may be harmful to patients with life-threatening infections. To overcome this limitation some authors have proposed to control the release of histones in NETs or lipids, or to develop harmless and more effective histone analogs. Moreover, in the future, a role for the epigenetic modification of histones could emerge [27].

Drotrecogin alfa (activated) is a recombinant human Activated Protein C (rhAPC), approved by FDA for the treatment of severe sepsis. In vitro data have shown that APC has antithrombotic, profibrinolytic, and anti-inflammatory effects [185]. Xu et al. demonstrated that the anti-inflammatory ability of APC, and its subsequent protective role against sepsis, is related to histone cleavage [26].

In a murine model of neonatal sepsis, Denning et al. reported the potential beneficial effect of the oligopeptide C23, derived from the protein CIRP, which is able to block the CIRP from binding its receptor. One hour after intraperitoneal injection of adult cecal material in C56BL/6 mouse pups, the mice received retro-orbital injection of C23 or of normal saline. Ten hours later, blood and tissues were analyzed. C23 treatment was able to more than halve the serum concentrations of pro-inflammatory cytokines IL-6 and IL-1β and significantly reduce the serum levels of AST and LDH collected for analysis. In the lungs, C23 treatment reduced expression of cytokines IL-6 and IL-1β by 78% and 74%, corresponding to a decrease in apoptosis and histologic lung injury score [92,186].

In murine models of thrombosis—including venous and arterial thrombosis—since it was reported that cfDNA and NETs contributed to the formation and stability of the thrombi, DNAses were administered in a mouse model of sepsis-induced intravascular coagulation, with subsequent clot lysis [97]. DNAses administration has also been found to be effective in a rat model of septic liver injury supported by Venoarterial Extracorporeal Membrane Oxygenation (ECMO). In such a model, DNase I significantly attenuated the level of alanine aminotransferase (ALT), aspartate aminotransferase (AST), NLRP3 inflammasome, myeloperoxidase (MPO), IL-1 β, and IL-18, and improved neutrophil infiltration [187]. Some authors have also proposed, in mice models of abdominal sepsis, a combination therapy of DNAses and low-molecular weight heparin (LMWH), but while the administration of either DNAse I or LMWH improved the survival of septic mice compared with saline, this was not true for combination-treated mice. These findings suggest that there may be a negative drug–drug interaction between DNAse I and LMWH [188].

NETs have been reported to be exuberant in sepsis; as such, the inhibition of their formation is a potential therapeutic target. Given the importance of the enzyme peptidyl arginine deaminase 4 (PAD4) in NETs formation, and the evidence that PAD4-deficient mice exhibit less organ damage and higher survival rate in a model of sepsis, it has been reported that the treatment of mice with CI-Amidine, a pharmacologic inhibitor of PAD4, decreases NETs formation and improves survival rates in sepsis [189,190].

Another proposed strategy aims to prevent the activation of mitochondria-related pathways through protecting the cardiolipin from oxidation with the administration of antioxidants like quinone-based antioxidants and long omega-3 polyunsaturated fatty acids (PUFAs), or the use of deformylase, the degrading enzyme for mtFPs [191,192].

With particular focus on inflammation-related hypercoagulability, some authors have proposed a role for the recombinant thrombomodulin (rTM). Thrombomodulin (TM), expressed on the endothelium, is crucial in regulating the coagulation system [193]. After an endothelial injury, TM is released into the intravascular space through cleavage. rTM seems to function as an inflammatory regulator, since it neutralizes DAMPs, including histones and HMGB1, and can directly and indirectly regulate NETs formation. For these reasons, it has shown good efficacy in suppressing inflammation in various experimental models, including sepsis [194]. In particular, a Phase 2b trial in 371 septic patients with suspected DIC showed lower mortality rates and sepsis-associated coagulopathy [195].

The role of alkaline phosphatase (AP) as a detoxifying enzyme has been investigated with regard to SA-AKI [193]. AP can remove one of the two phosphate groups of the lipid A of the endotoxin, and even if the dephosphorylated endotoxin can still bind to TLR4, it is no longer able to activate the receptor, showing TLR4-antagonistic effects. In vitro and in vivo studies on murine models have revealed the presence of phosphatase activity at the tubular brush border, confirming the protective ability of AP in Gram-negative *Escherichia coli* sepsis animal models, but not in Gram-positive sepsis sustained by *Staphylococcus aureus.* Furthermore, AP catalyzes the dephosphorylation of eATP to its metabolites (ADP and AMP). Binding to its receptors in the nephron, adenosine seems to act on the renal vascular tone, on the regulation of the glomerular filtration rate, and on the renin release [196]. Given these observations, and after the evidence of good results in animal models of sepsis, some authors investigated the effects of bovine intestinal AP (biAP) administered intravenously to suspected or proven Gram-negative septic patients (with or without AKI). No safety concerns emerged, and in the treated arm, an improvement of endogenous creatinine clearance (ECC) was observed, with respect to the placebo arm [197].

Moreover, because of its role in SA-AKI, TLR4 has been investigate as a target of antibodies, peptides, nanoparticles, lipid A analogs, and derivatives of natural products [197,198].

In addition, since 1990, some synthetic anti-lipopolysaccharide peptides (SALPs) have been objects of research. These peptides can bind and inhibit not only the LPS from Gram-negative bacteria but also lipoproteins (LP) of Gram-positive origin. The effect of this binding is—besides its antibacterial action, regardless of the bacterial resistance—the prevention of uncontrolled inflammation [199]. LPS binders attack both wall and free endotoxin, targeting one of the three different parts of the LPS (O-antigen, core, Lipid A) and having various compositions directed to different classes of bacteria. This implies that the limit of these molecules is the LPS heterogeneity. Therefore, it has been proposed to search for well-conserved parts of the endotoxin, and a promising candidate could be part of the inner core region called 3-deoxy-D-manno-octulosonic acid (KDO) [45], or the lipid A with fully synthetic disaccharide lipid A mimetics (DLAMs) [46]. Among these peptides, we report, as an example, the LPS-binding peptide 19–2.5, which not only interferes with the activation of the coagulation and contact system but also prevent the interaction of LPS with the high molecular weight kininogen (HK), one of the major LPS carriers in blood [200].

Interestingly, in a *P. aeruginosa* infection in a mouse burn model, the combination of prophylaxis (pre-infection) and therapeutic (post-infection) treatment with anti-flagellin sub-type monoclonal antibody (anti-fla-a) limited the bacterial dissemination and invasiveness, with subsequent reduction of septic mortality and morbidity [201]. Similar results have been reported by other investigators [202,203].

Besides DAMPs and PAMPs modulation, another promising field of research is that of receptor modulation.

TLR4 is among the most studied receptors. It is of no surprise that some natural inhibitors of this receptor have been detected in the past years. Those act at various levels: extracellular (i.e., soluble CD14, soluble MD2, and soluble TLR4), cell membrane (e.g., receptor RP105, TNF-Related Apoptosis Inducing Ligand-Receptor TRAIL-R, and receptor ST2), and intracellular (e.g., short form of MyD88, interleukin-1 receptor associated kinase M IRAK M, Toll Interacting Protein TOLLIP, β-arrestin, Suppressor of cytokine secretion 1 SOCS1, and others). Also, some drugs have been investigated for their ability in modulating TLR4 through various mechanisms. Among molecules interfering with TLR4 and its mRNA expression, there are many used for several applications, like chloroquine [204], ketamine [205,206,207], statins [208,209], and lidocaine [210]. Among drugs acting on the TLR4 transduction pathway, there are eritoran (E5564), resatorvid (TAK242), ketamine, opioids [211], vitamin D3, and lansoprazole. Furthermore, some forms of anti-TLR4 and anti-TLR4-MD2 complex have been texted on cultured cells and in animal models exposed to LPS [212].

Once again, most of the illustrated agents have been tested in vitro or in vivo in animal (mostly) murine models, and not all of them in the context of sepsis. Moreover, there are contrasting results on their effect on TLR4 functions; for example, some authors have reported that vitamin D3 increases the expression of the receptor in tuberculosis spondylitis patients [213], while others have suggested that its supplementation inhibits the TLR4/MyD88/NF-κB signaling [214]. It is interesting to remark that two cited molecules, both antagonists of TLR4, the synthetic lipid A antagonist eritoran and the small molecule resatorvid, reached phase III clinical trials as antisepsis agents, but both (involving 1961 and 274 patients, respectively) failed to meet their endpoints [215,216]. Nowadays, Monophosphoryl Lipid A (MPLA) is the only TLR4 agonist to be approved by the FDA for use as a vaccine adjuvant in humans (Cervarix R ©, Fendrix R ©) [217], and its effectiveness in sepsis is under investigation [218,219]. Vega et al. explored its potential as an immunomodulator. In fact, the molecule is a synthetic and detoxified form of the endotoxin LPS, and when administered to septic patients, there is a weaker induction of target genes compared to that induced by LPS stimulation. This effect may depend on the fact that MPLA-induced TLR4 transduction pathway is preferentially non-MyD88-mediated, following the TRIF-dependent cascade [219,220]. With regard to other TLRs immune modulators [221], they are approved medications or still objects of trials in different phases of development for various conditions, but we have not found available past studies or current clinical trials for their application in sepsis. Moreover, the TLRs transduction pathway could be modulated at the effector level. As an example, some authors have developed small molecules that target the BB-loop region of MyD88, which is essential for the homo (adaptor–adaptor)- and hetero (receptor–adaptor)-dimerization that is necessary for the function of TIR domains of TLRs. Most of the studies have been conducted in animal models of inflammatory diseases, but the results suggest that these molecules might have a future role in the treatment of sepsis [222].

As RAGE have been reported to be engaged in sepsis pathogenesis, several strategies have been used to study the effects of RAGE inhibition, such as antibodies against RAGE, RAGE knockout mice, siRNA to silence RAGE, and soluble RAGE. These studies have showed that RAGE inhibition limits the release of pro-inflammatory cytokines but also interferes with phagocytic clearance of apoptotic lymphocytes, thus protecting against sepsis-induced increases in endothelial permeability and hypercoagulability. Based on these observations, the inhibition of RAGE pathways has been proposed as a therapeutic tool against sepsis, the functional outcomes of which may rely on the degree of RAGE expression in specific tissues, the characteristics of each ligand, and its interaction with different extracellular domains of the receptor. Unfortunately, experimental results have yielded conflicting results, so that further studies are required to better define the role of RAGE in sepsis and its treatment [73,223,224,225].

Among soluble receptors, we also mention the synthetic sTREM-1 antagonistic peptide, nangibotide. It inhibits the activation of the TREM-1 receptor, thus decreasing leukocyte activation and innate immune response [226], with subsequent protective effects on the cardiovascular system and survival [227]. The medication has been demonstrated to have a good safety profile and promising effects on sepsis morbidity and mortality [228,229].

Since the uncontrolled activation of NLRs (mainly NLPR3) and GSDMD have been described both in human and mouse models of sepsis, caspase 1/4/5/11 have been proposed as drug targets. Some inhibitors of this pathway are unsuitable in clinical practice since they act like “paninhibitors”, blocking not only pyroptosis but also apoptosis. In contrast, selective inhibitors have shown promising effects in vitro and in vivo, treating inflammatory diseases of mice, but unfortunately, little clinical research has been conducted in human inflammatory diseases [61]. Studies have also suggested that the administration of inhibitors of NLRP3/caspase-1 pathway-mediated pyroptosis (e.g., MC950, melatonin) is protective against sepsis-associated encephalopathy, spinal cord injury, and motor neuron damage in rats [60].

Blocking P2 × 7 receptor with AZ 10,606,120 exacerbates vascular hyperpermeability and inflammation in murine polymicrobial sepsis [109]. In addition, in mouse models of CLP-induced sepsis, the targeting of P2 receptors has been suggested as a modulator of PMNs responses. In fact, the ATP receptor antagonist suramin diminished CD11b expression and subsequent PMNs activation [230].

## 3. Findings, Open Problems, and Future Perspectives

Sepsis is a complex and kaleidoscopic syndrome. Since the paradigm about its pathogenesis has been switched from a pathogen-centered to a pathogen–host-mediated one, many features have been elucidated. It has become clearer that, given the multiple and dynamic possible combinations of pathogen–host interaction, not only is every patient with sepsis unique but the same patients may also express several clinical manifestations at different moments of the disease history. However, even if physicians are aware that every septic patient is inimitable in their clinical features, there is the need to sort patients into macro-categories to manage them clinically and therapeutically in an evidence-based fashion. For these reasons, during recent years, many efforts have been made to identify sepsis phenotypes. For example, Seymour et al. identified four sepsis phenotypes with different clinical and biochemical features. The α phenotype had fewer abnormal laboratory values and less organ dysfunction. Patients with the β phenotype were older, had greater chronic comorbidities, and had higher risk of presenting with renal dysfunction. Those with the γ phenotype were more likely to have fever, high white blood cell count, higher erythrocyte sedimentation rate, or C-reactive protein. Finally, those with the δ phenotype had hypotension, elevated serum lactate levels, and high transaminase levels [231]. Yet, there is no classification of sepsis based on the expression of DAMPs or PAMPs, even if, from our review of the literature, it emerges that there might be differences in their concentration depending on the pathogen, the source of infection, the damaged organs, or the moment of their measurement.

From our examination of the available scientific works on the topic, two main fundamental findings emerged. First, we believe that—despite the enthusiasm showing through some papers—there is not sufficient evidence to define molecular patterns (MPs) as biomarkers for imminent clinical use in human sepsis, since we have not been able to find works reporting data of their normal and sepsis-related cut-off values, nor of their sensitivity and specificity in the diagnosis of sepsis. There is emerging literature about their relationship between the severity of sepsis and prognosis, but most of the studies have been conducted on animal models, in which sepsis features are not completely analog to the human ones. Second, there is no study which systematically assessed the trend of the concentration of PAMPs and DAMPs in the progression of sepsis. However, bringing together the information of various works, we tried to reconstruct the existence of a temporal line of DAMPs and PAMPs release during sepsis (Figure 6). In a murine model of sepsis induced by *E. coli* injection, cfDNA showed an increase during the first hours of the disease, reaching a 20-fold increases at 5 h after sepsis induction [130]. This increase in cfDNA seems to also be explained by low Deoxyribonucleases (DNases) activity during sepsis. However, the lack of standard values in DNase activity makes the comparison of different studies quite difficult [232] Sumi et al. reported that plasma eATP, and its metabolites adenosine diphosphate (ADP) and adenosine monophosphate (AMP), concentrations increased up to 6-fold during the first 8 h after CLP [230]. HMGB1 levels are significantly elevated at the later time point of sepsis, reaching a plateau at 16–32 h after the onset of sepsis in both animal models of sepsis and septic patients, and its circulating levels remain elevated for weeks [233,234]. In a mouse model, marked eCIRP elevation was observed at 20 h after sepsis; it continued to increase at 48 h, while the highest increase seemed to occur at 72 h after the onset of sepsis, suggesting that this rise can be clinically associated with late-stage immunosuppression in sepsis [91]. With regard to mtDNA it was reported to increase on day 1 and remain constant until day 5 in patients with sepsis [136]. We are aware that this reconstruction has many limitations, since the data have been taken from studies using different sepsis models—mostly murine models—and measurement techniques on different samples; moreover, the underlying mechanism of MPs release has not been elucidated in the studies, and it is known that different types of cell death and active release pathways may affect the type of DAMPs found in the circulation [16]. Moreover, while, for some DAMPs, the trend has been quite precisely described, even with referral to the folds of concentration increase from the baseline, reporting its modification not only in the first hours but also in the first days after sepsis onset, this information is lacking for most of them. We believe that—despite the previously exposed limitations—the use of animal models is pivotal for the purpose of studying the temporal trend of MPs in sepsis. In fact, it could be useful to define this trend in the absence of antibiotic therapy (which is unethical in humans) and after their administration, to understand how antimicrobial agents affect the concentration of these molecules (e.g., differences between bacteriostatic and bacteriolytic agents). In humans, it should be considered that septic patients are often elderly, multi-morbid, in polytherapy, and managed in intensive of sub-intensive settings. Nevertheless, even if there is evidence that some commonly used drugs interfere with the release of DAMPs, there is scarce literature on the interaction among alarmins and these drugs, which may significantly affect the natural history of sepsis. It is our opinion that further research on this topic could be of great clinical significance since it could help in the future definition of the timeline of sepsis, which could be of great importance for prognostic and therapeutic considerations and decision-making.

Another important observation emerging in our paper is that not all the proposed therapeutic strategies addressed to MPs modulation have been specifically studied in human sepsis, since some of them have only been studied in vitro or animal models, often with small sample sizes. Once again, even if these studies offer some remarkable data, we should limit the enthusiasm. For example, some approved EBPs techniques have been demonstrated to have inconclusive effect in reducing sepsis mortality. Data on one of the most promising EBPs—i.e., hemoadsorption—are limited to some DAMPs (i.e., histones and HMGB1), and we have not been able to find studies evaluating the potential removal of other DAMPs. Ideally, a better comprehension of the single patient’s molecular profile could be targeted in the future with tailored absorption cartridges or, as some other authors are starting to test, with nanotechnologies, but again, the current evidence is not enough to systemically recommend these strategies in septic patients. On the contrary, there is evidence that EPBs might induce serious complications like severe dyselectrolytemias, arrhythmic events, and hemodynamic instability, and interfere with the antibiotic therapy itself, which is, to date, the only universally accepted treatment for sepsis [162]. Several questions remain unanswered regarding the scheduling, interval, and frequency of administration and the optimal membrane characteristics (molecular weight cut-off, surface area, composition) for EPBs in septic patients [161]. Regarding other strategies, such as the administration of small molecules, antibodies, and vaccines, we again underline the limits of the use of animal models, mainly of a methodological nature. In facts, some studies evaluate the efficacy of some agents as more than as therapeutic agents, such as as prophylaxis, through their administration before the septic challenge. This strategy, which can be of some usefulness in understanding the mechanism of effect of the agent, rarely finds its application in real life since we cannot predict the development of sepsis but only—and in limited cases—estimate the risk of its development (e.g., patients exposed to invasive procedures, particular categories of patients like those with indwelling devices or with pre-existing conditions affecting the immune system like diabetes of immunodeficiency). Given these considerations, to date, only a few of these strategies have been approved by some drug regulatory agencies, and the available evidence is too limited to systematically insert them in the standard therapy of sepsis. Additionally, since most of the studies were conducted on animal models which are euthanized at predetermined moments of the experiment, there are scarce data on the long-term effects of MPs modulation, a topic that might be of crucial importance since it is known that MPs are involved in the pathogenesis of autoimmune diseases and tumors. In fact, the scientific community is discussing the need to identify and define an “homeostatic window” of DAMPs and SAMPs (suppressing/inhibiting molecular patterns) concentrations to guarantee a safe treatment modality for patients [180]. In the light of this analysis, it is our opinion that besides the enthusiasm, prudence is mandatory, and that future studies on such strategies are necessary to better understand the role of innate immunity on sepsis and provide other therapeutic tools.

## 4. Materials and Methods

This review consists of two research phases. In the first phase, we included studies, published in English over the last 5 years, on the topics of DAMPs and PAMPs in sepsis. We searched on PubMed^®^ and Cochrane^®^. The keywords we searched for were “DAMPs OR PAMPs AND sepsis OR septic”. The inclusion and exclusion criteria are reported in Table 6. All the articles were read, and their bibliographies were checked to select other reputed and relevant works based on the opinion of the authors. In the second phase, we focused on the research, with the same criteria, of evidence regarding PAMPs and DAMPs in the single microareas explored in the review (e.g., “DAMPs OR PAMPs AND endothelium). No ethical approval was required to perform this review.

## 5. Conclusions

In the recent pandemic years—more than ever—sepsis has been an hot topic, and the role of DAMPs and PAMPs as biomarkers, prognostic factors, and potential therapeutic targets has gained crucial attention [235,236]. The literature on the topic is proliferating, demonstrating the great effort in finding new, faster, and more effective ways of detecting, classifying, and managing sepsis with strategies complementary to the conventional therapy. Unfortunately, besides a few approved new strategies, many of them are far from being applied in clinical practice. Several points need to be elucidated since we may affirm that many DAMPs and PAMPs express both pro-inflammatory and anti-inflammatory functions, depending on their localization in the cell (nuclear, cytosol, membranes), in different cells (immune and non-immune cells), and even in different phases of sepsis. This evidence might be the starting point for deeper knowledge of the different phenotypes of sepsis and more rational use of the actual therapeutic options—for example, different strategies at different moments of sepsis—and the study of new treatment strategies.

## Figures and Tables

**Figure 1 ijms-25-00962-f001:**
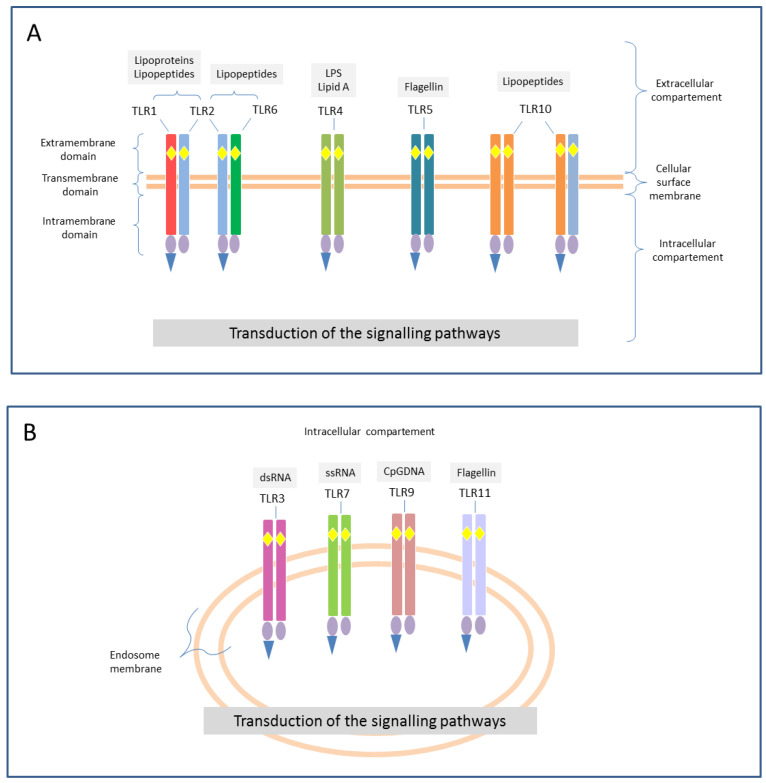
Examples of TLRs’ morphology. Toll-like receptors (TLR) are membrane receptors, found both in the cellular surface (square **A**) where they can exist as hetero- or homodimers (TLR1, 2, 4, 5, 6, and 10), and in the intracellular compartments (square **B**) as homodimers (TLR3, 7, 8, 9, and 11). The different colors indicates the different structure of each receptor, and their possible combinations (e.g., the same TLR2, reported in light blue can form heterodimers with TLR1, TLR6 and TLR10 as well). They are type I transmembrane glycoproteins and share a common structure: extramembrane domain that includes the leucine-rich repeats (LRRs, yellow rhombuses) with ligand-binding function; transmembrane (or intermediate domain); intramembrane domain that includes the same Toll/IL-1R (TIR, violet ovals) domain as IL-1R, which plays a role in signal transduction through the reclamation of several adaptor molecules (blue triangle). Depending on the nature of these adaptors, TLRs signaling can be classified into myeloid differentiation factor 88 (MyD88)-dependent and MyD88-independent pathways. Abbreviations: CpG-DNA = unmethylated cytosine-phosphate-guanine DNA; dsRNA = double stranded RNA; LPS = lipopolysaccharide; RNA = ribonucleic acid; ssRNA = single stranded RNA.

**Figure 2 ijms-25-00962-f002:**
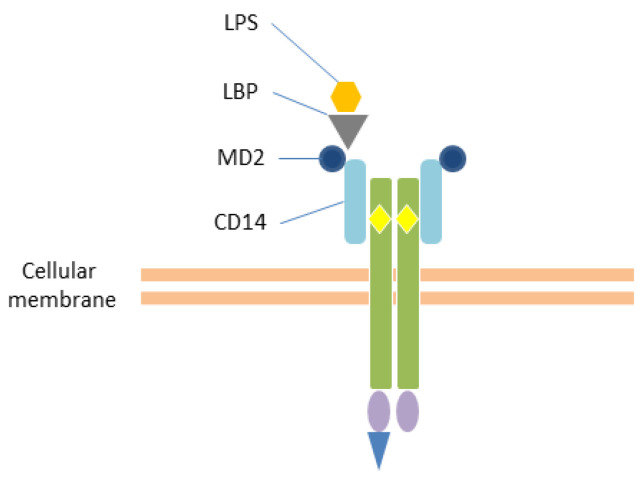
Interaction with LPS. TLR4 recognizes the lipopolysaccharide (LPS) in association with the myeloid differentiation factor 2 (MD2) and the LRR structural protein CD14. The LBP (LPS-binding protein) transports the LPS to the CD14 (cluster of differentiation 14) on the cell membrane of monocytes and macrophages. This interaction further promotes the heterodimerization of TLR4 and subsequent signaling. Yellow rhombuses: Leucine Rich Repeats (LRRs); violet ovals: Toll/IL-1 receptor (TIR) domain; blue triangle: adaptor molecule.

**Figure 3 ijms-25-00962-f003:**
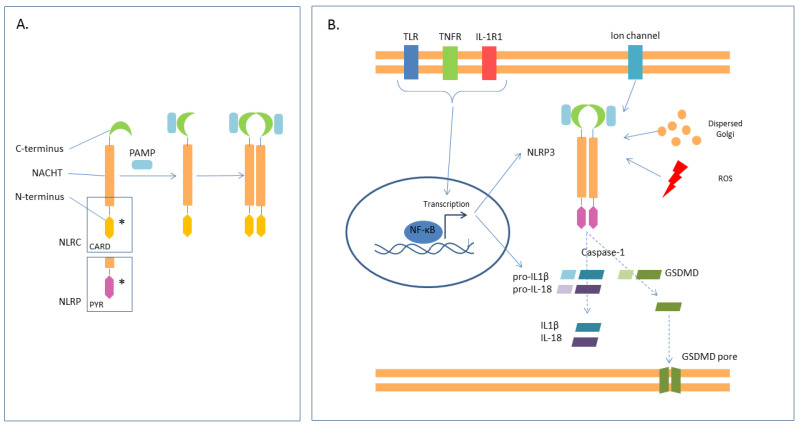
NLRs—(**A**). NLRs have a central, nucleotide-binding domain (NACHT), a C-terminus that identifies and binds the ligand, and a N-terminus with effector function which could be a CARD (caspase activation and recruitment domain) or PYR (pyrin domain) (*). The interaction with the ligand causes conformational change and oligomerization. The most characterized NLR is NLRP3. (**B**) NLRP3 is required for the activation of the canonical inflammosome. The first signal (priming) is the NF-kB mediated transcription of the NLRP3 gene together with pro-IL-1B and pro-IL-18. This first step may be stimulated by activation of TLRs, TNFR, and IL-1R. The second signal is mediated either by PAMPs or by transmembrane ionic fluxes, reactive oxygen species, and Golgi dispersion. This signal causes oligomerization, with activation of caspase-1, that cleaves the pro-interleukins in mature IL-1 and IL-18, and the gasdermin (GDSMD), which then forms membrane pores.

**Figure 4 ijms-25-00962-f004:**
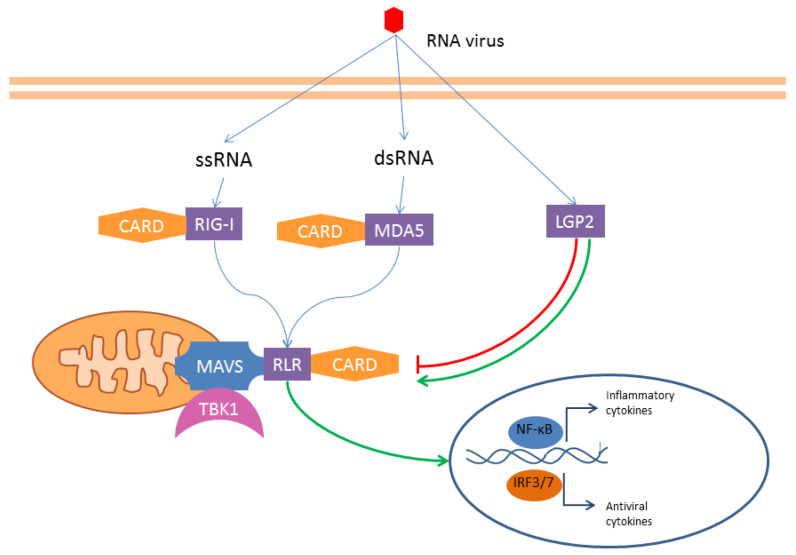
RLRs—The RNA helicases RIG-I, MDA5, and LGP2 interact with viral RNA (and eventually with DAMPs). While LGP2 has a regulatory function, RIG-I and MDA5 have a CARD domain. These two latter receptors interact with a mitochondrial transmembrane adaptor (MAVS) that binds the kinase TBK1 (TANK-binding kinase 1). The downstream pathway leads to inflammatory and antiviral cytokines transcription release.

**Figure 5 ijms-25-00962-f005:**
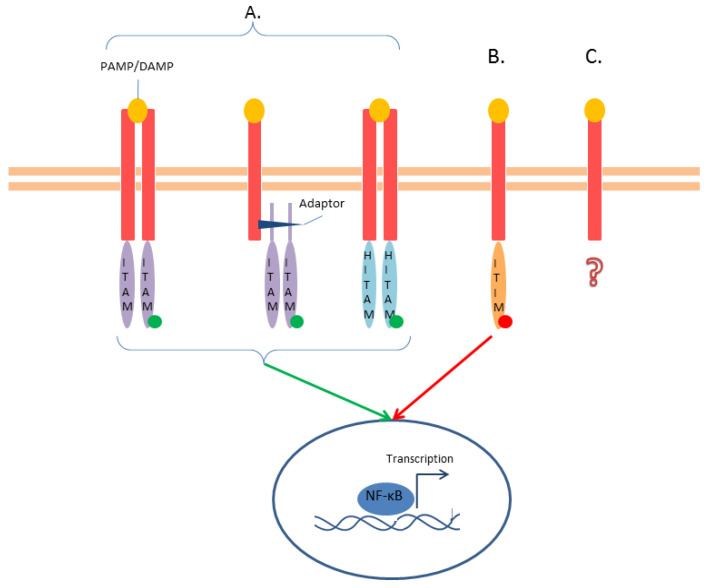
CLRs—(**A**). Some CLRs have activating function (green arrow) thanks to their activating ITAM domain. This domain can be included in the intracellular tail, connected to the receptor by an adaptor, or be half-domain (HITAM). The phosphorylation of the tyrosine (green dot) causes a downstream signaling that leads to NF-κB pathway activation. (**B**) Other CLRs contain an inhibitory motif (ITIM) that recruits tyrosine phosphatases (red dot) to inhibit the signaling pathways (red arrow). (**C**) Some CLRs have neither ITAM nor ITIM domains, and their signaling is either uncharacterized or utilizes alternative pathways (question mark).

**Figure 6 ijms-25-00962-f006:**
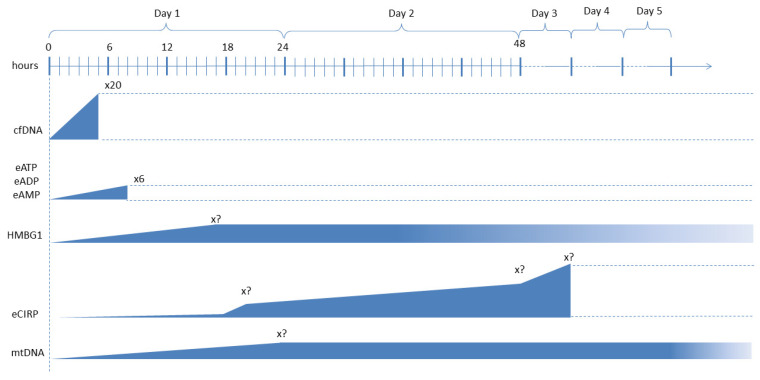
Temporal increase trend of some DAMPs during the first phases of sepsis—Note that, for the DAMPs for which a precise increase trend (folds from the baseline) has been reported, it has been indicated in the figure in an in-scale manner (see cfDNA, eATP, and its metabolites). For the other DAMPs, the lack of this information has been identified with “x?”. Moreover, we highlight that while, for some DAMPs, the trend even after the first days from sepsis (i.e., HMGB1, mtDNA) has been described, for others, this information is missing (dotted lines). For these reasons, the study of the precise concentration trends of DAMPs in sepsis could be an interesting field of research that could have potential important clinical implications.

**Table 1 ijms-25-00962-t001:** Overview of the most studied DAMPs.

Type of DAMP	Normal Location and Function	Mechanism of Release		Receptors		Pro-Inflammatory	Anti-Inflammatory
		Passive	Active	PRR	Non-PRR		
HMGB1	Ubiquitous, nuclear, non-histone chromatine-binding protein. Transcriptional activity of various proteins, DNA replication, DNA repair, and nucleosome formation.	+	+	TLR2 TLR4 TLR9	RAGE TREM-1	reduced form, disulfide-bond possessing form	oxidizing form
CIRP	Nuclear, RNA chaperone protein. Cell proliferation, survival, and tumor formation and progression.	−	+	TLR4 IL-6R NLRP3	TREM-1	Inflammosome induction NETs formation	IL-6 binding induces STAT3 that enables the NF-κB pathway
Histones	Nuclear, support the normal chromatine structure, regulate gene expression.	+	+	TLRs		TLR-mediated cytokine release and membrane permeability impairment	Dysregulation of methylation, lactylation, and citrullination cause.
ATP	Ubiquitous, intracellular source of energy.	+	++		P2X	High levels → assembly of the NLRP3 inflammosome.	Low levels → inhibition of macrophagic IL-2 and TNF-α secretion; metabolites ADP and AMP.
NAD	Intracellular and extracellular metabolite, and cofactor to many enzymes.	+	+		P2X, P2Y	Inflammosome assembly.	
HSP	Molecular chaperones.	+	+	TLR2 TLR4 CLRs		eHSP-60 and eHSP-70	eHSP-70 eHSP-27
RNA	Intracellular.	+	++	TLR3 (dsRNAs) TLR 8 (ssRNAs) RIG-1 (dsRNAs)			
DNA	Nuclear.	+	+	TLR9 AIM	RAGE		
mtDNA	Mitochondrial.	+	−	TLR9 NLRP3		Inflammosome assembly.	
mtFPs	Mitochondrial.	+	−		FPRs	Chemiotaxis Oxidative burst	

Abbreviations: AIM = absent in melanoma; AMP = adenosine monophosphate; ADP = adenosine diphosphate; ATP = adenosine triphosphate; CIRP = cold-inducible RNA-binding protein; CLRs = c-type lectin receptors; DAMP = damage associated molecular pattern molecules; DNA = deoxyribonucleic acid; dsRNA = double stranded RNA; FPRs = formyl peptides receptors; HMGB1 = high mobility group box 1; HSP = heat shock proteins; IL = interleukin; mtDNA = mitochondrial DNA; mtFPs = mitochondrial N-formyl peptides; NAD = nicotinamide adenine dinucleotide; NETs = neutrophil extracellular traps; NF-κB = nuclear factor κB; NLRP3 = nucleotide-binding oligomerization domain-like receptor (NLR) family pyrin domain containing 3; RAGE = receptor for advanced glycation end products; RIG-1 = retinoic acid-inducible gene 1; RNA = ribonucleic acid; ssRNA = single stranded RNA; STAT3 = signal transducer and activator of transcription factor 3; TLR = Toll-Like Receptors; TNF = tumor necrosis factor; TREM-1 = triggering preceptors expressed on myeloid cells-1.

**Table 2 ijms-25-00962-t002:** Overview of the human TLR.

Type TLR	Normal Location	PAMP Ligands	DAMP Ligand
TLR1	Cell surface	Lipoprotein and LTAs	Unknown
TLR2	Cell surface	Lipoprotein and LTAs	HMGB1, HSP60, HSP70, HA
TLR3	Intracellular (endosomal)	ds-RNA	mRNA
TLR4	Cell surface	LPS	HSP60, HSP70, HA, HMGB1, fibrinogen, histones
TLR5	Cell surface	Flagellin	HMGB1
TLR6	Cell surface	Lipoprotein and LTAs	Amyloid β, oxidized LDLs
TLR7	Intracellular (endosomal)	ss-RNA	ss-RNA
TLR8	Intracellular (endosomal)	ss-RNA	ss-RNA
TLR9	Intracellular (endosomal)	CpG-DNA	cfDNA, mtDNA, histones
TLR10	Cell surface	Diacylated lipopeptides	Unknown
TLR11	Intracellular (endosomal)	Flagellin, profiling	Unknown

Abbreviations: PAMP = pathogen-associated molecular pattern molecules; DAMP = damage-associated molecular pattern molecules; cfDNA = cell free DNA; CpG-DNA = unmethylated cytosine-phosphate-guanine DNA; mtDNA = mitochondrial DNA; HA = hyaluronic acid; HMGB1 = high mobility group box 1; HSP = heat shock protein; mRNA = microRNA; LDLs = low density lipoproteins; LPS = lipopolysaccharide; LTAs = lipoteichoic acids; ss-RNA = single-stranded RNA, TLR = tool like receptor.

**Table 3 ijms-25-00962-t003:** Tissue-specific expression of TLRs-mRNA in human tissues.

Type TLR	Predominant Concentrations Detected in:
TLR1	Kidneys, lungs, spleen
TLR2	Lungs, spleen, brain, heart, and muscle
TLR3	Placenta
TLR4	Spleen
TLR5	Ubiquitously
TLR6	Ubiquitously
TLR7	Lungs, placenta, spinal cord, spleen
TLR8	Lungs, spleen
TLR9	Skeletal muscle, spleen
TLR10	Spleen, thymus

**Table 4 ijms-25-00962-t004:** Examples of studies reporting the main observations regarding DAMPs and PAMPs in those organs and systems included in the evaluation of SOFA score.

	Respiratory	Nervous System	Cardio- Vascular	Hepatobiliary	Coagulation and Endothelium	Kidney
HMGB1			HMGB1 are among the most cardiotoxic DAMPs in sepsis [83]	HMGB1 levels increase during acute liver failure [84] Their release by the hepatocytes is promoted by LPS in a caspase1-/gasdermin-mediated pathway [22]	HGBM1 can bind to ECs and dose-dependently upregulate the expression of adhesion molecules, the production of pro-inflammatory cytokines, and the hyperpermeability [13,22] HGBM1 is essential for platelet activation and degranulation, contributing to hemostasis and NETs formation [22]	HMGB1—depending on its oxidized/reduced abundance—has been proven to induce a phenotypic of tubular epithelial cells in pro-inflammatory cells during sepsis [85]
HISTONES	Hemorrhagic, thrombotic, and fibrotic phenomena in the alveoli and septa of lungs of mice challenged with lethal doses of histones [26] The lungs of children with ARDS and sepsis showed higher levels of some histones when compared to only septic children [86]		Histones are among the most cardiotoxic DAMPs in sepsis [83]		Histones are cytotoxic to the endothelium in a dose-dependent fashion, inducing expression of adhesion molecules, oxidative stress, pyroptosis, and shedding of the glycocalix, with imbalance between procoagulant and anti-coagulant factors, which could lead to DIC [84,87,88] Cell free histones—in a TLR2 and 4-mediated mechanism—activate platelets, with subsequent aggregation, chemokine secretion, and thrombin formation [84,89] Megakaryocytes contain extranuclear histones that are transferred to their platelet progeny, where they are found in high concentrations in septic patients [90]	
eCIRP	Induces the NLRP3 inflammosome in macrophages, which has been related to ALI and subsequent ARDS in sepsis [91] Its concentration has been independently associated with severe hypoxia and higher complexity of respiratory support during COVID-19 lung failure [92]				eCIRP promotes the endothelial dysfunction via a NLRP3-mediated pathway [93]	
cfDNA				Higher levels in patients who underwent a liver transplant correlate with higher risk of death for liver abscess [94]	NETs are enhanced by activated platelets, contributing to the clearance of pathogens [95,96,97] cfDNA impairs the expression of endogenous anticoagulant agents like the protein C [98]	NETs may drive an evolution of kidney macrophages in a M2 phenotype that promotes tubular regeneration after septic harm [85]
eATP and metabolites						Adenosine may drive an evolution of kidney macrophages in a M2 phenotype that promotes tubular regeneration after septic harm [85]
mtDNA	Higher concentration correlates with higher vascular permeability, accumulation of TNF and IL-6 in lung lavage fluids, and PMN infiltration in the airways [39]					
mtFP					ECs express FPRs that could be stimulated by mtFPs, with subsequent increased endothelium permeability and tissue hypoperfusion [39]	
LPS			Detrimental effects on contractility [99]		Promotes the expression of coagulation factor III, the main activator of septic coagulopathy, on platelets, ECs, and monocytes	
TLRs		Expressed neurons [17]	The co-culture of macrophages and cardiomyocytes exposed to LPS show the activation of TLR3 and TLR4, with subsequent activation of NF-κB-mediated transcription. The experimental inhibition of TLR4 prevents these events [100] Implicated in the impaired calcium storage in the sarcoplasmic reticulum [101]		Several TRLs are expressed in the ECs, with different tissue distributions that may explain the various effects of sepsis in different organs. TLR1 and 4 are found in the umbilical cord ECs, and TLR3 and 9 in the human aortic ECs [102] TLRs are expressed in platelets [89] TLR4 expressed in platelets can be considered as a bridge connecting thrombosis and innate immunity [103]	TLR4 is activated by the urinary DAMP uromodulin, also known as Tamm-Horsfall protein, which is non-immunogenic inside the tubular lumen but becomes immunogenic after cell damage and translocation in the interstitial compartment [85] TLR4 expression is low in the renal medulla and high in the cortex. Its expression increases during ischemia/reperfusion and sepsis [104]
NLRs	NLRC3 negatively regulates NF-κB during the progression of sepsis-induced ALI [105]		NLRP3 inflammasome is activated in cardiac fibroblasts during sepsis [60]		eCIRP promotes the endothelial dysfunction via a NLRP3-mediated pathway [93] NLRs are expressed in platelets, and the NLRP3 inflammosome has pro-thrombotic functions [89] NLRP3 triggers a gasdermin-mediated release of coagulation factor III [106,107]	
CLRs					CLRs are expressed in platelets [89] The stimulation of CLRs—like CLEC2—by viral particles and mtDNA promotes platelet activation, degranulation, and thrombus formation [69]	
RAGE				RAGE-mediated signaling has been associated with acute inflammatory liver stress that can contribute to multiorgan failure. Rage −/− mice show lower levels of hepatic pro-inflammatory cytokines and DAMPs [108]		
P2					Blocking the P2X7R with a synthetic antagonist in rats with sepsis reduces the endothelial dysfunction [109]	

Abbreviations: ALI—Acute Lung Injury; ARDS—Acute Respiratory Distress Syndrome; DIC—Disseminated Intravascular Coagulation; ECs—Endothelial Cells; NETs—Neutrophil Extracellular Traps.

**Table 5 ijms-25-00962-t005:** Examples of “selective” therapeutic strategies targeting DAMPs, PAMPs and their signaling pathways.

	Therapeutic Strategy
Type of DAMP	
HMGB1	- HMGB1 neutralizing antibodies - Heparin, heparin variants, or heparinoids - Recombinant HMGB1 A box, P5779 peptide derived by the HMGB1 box B
CIRP	- Oligopeptide C23
Histones	- Histones and their fragments - Drotrecogin alfa (activated)
ATP, ADP, AMP	- Alkaline phosphatase
DNA	- DNAses
mtDAMPs	- Quinone-based antioxidants - Long omega-3 polyunsaturated fatty acids (PUFAs) - Deformylase
Type of PAMP	
LPS	- Alkaline phosphatase - Synthetic anti-lipopolysaccharide peptides (SALPs)
Flagellin	- Anti-flagellin sub-type monoclonal antibody
Type of Receptor	
TLR4	- Alkaline phosphatase (AP) - Eritoran - Resatorvid (TAK242) - Small agonists/antagonists (e.g., MPLA)
NLRP3	- MC950 - Melatonin
P2	- AZ 10606120 - Suramin
RAGE	- Anti-RAGE antibodies - siRNAs - Soluble RAGE
TREM-1	- Soluble TREM-1 (Nangibotide)

**Table 6 ijms-25-00962-t006:** Inclusion and exclusion criteria.

Inclusion Criteria	Exclusion Criteria
Type of article: in order of importance, we considered clinical trials, observational studies, systematic reviews, ad narrative reviews.	Topic: articles treating other unrelated topics.
Language: only articles written in English.	
Year of publication: articles written in the last 5 years.	

## Data Availability

Not applicable.

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
