# Peer review of "PAMPs and DAMPs in Sepsis: A Review of Their Molecular Features and Potential Clinical Implications"

_ijms, 2024, doi:10.3390/ijms25020962_

Round 1
Reviewer 1 Report
Comments and Suggestions for Authors
I have read the review manuscript „PAMPs and DAMPs in sepsis: a review of their molecular features and potential clinical implications“ written by Cicchinelli S et al.
I think it is not necessary to mention the inclusion of „Uptodate“ articles since the authors mentioned the articles searched on Pubmed and Cochrane.
It would be good to put the burden of sepsis in perspective, i.e. to mention incidence and mortality numbers, complications etc. I don't think there is need to mention terms like bioenergetics.
It is unclear what is the purpose and explanation for differentiating between PRRs (pattern-recognition receptors) and non-PRRs.
Regarding the part of the manuscript describing PAMPs and DAMPs as potential biomarkers, the authors 55.3% is good sensitivity for plasma HMGB1 in predicting multiorgan failure in septic pediatric patients, which is a problematic statement. Also, it would be logical to mentioned the main studied non-PAMP/DAMP biomarkers for sepsis so far.
I find the whole part of the manuscript describing treatment strategies problematic. Main treatment options are based on removal of PAMPs and DAMPs which is an oversimplistic view; there is no mention of potentially deleterious consequences of PAMP, DAMP, cytokine removal or neutralization on effective immunologic response and resolution of infection. Likewise, there is no commentary on the effect of “removal technologies” on antibiotic pharmacokinetics. There is even mention of “moderate antibiotic treatment” and septic molecules removal combination resulting in 100% survival of septic mice, which is not further explained. Disregarding this non-comprehensive part of the manuscript, most of the manuscript is written like a chapter from an immunology book, lacking novel, clinically meaningful data.
The manuscript needs to undergo extensive grammatic and syntactic corrections.
Some examples:
- starting a sentence with „Anyway,“, or
- „Cell swelling and plasma membrane rupture, under extreme chemical or physical insults such as the presence of toxins or trauma, characterize it.“ or
- „Human Immunodeficience Virus“ or
- „It is well known that sepsis is a deregulated host response to infection, and in the previous paragraphs, it resulted evident that the first deregulations involve the immune cells.“
Comments on the Quality of English LanguageModerate editing of English language required.
Author Response
Thanks for the time the reviewer spent evaluating our article and for the suggestion that made us able to significantly improve our paper.
I have read the review manuscript „PAMPs and DAMPs in sepsis: a review of their molecular features and potential clinical implications“ written by Cicchinelli S et al.
I think it is not necessary to mention the inclusion of „Uptodate“ articles since the authors mentioned the articles searched on Pubmed and Cochrane.
Answer: we deleted the mention to Uptodate
It would be good to put the burden of sepsis in perspective, i.e. to mention incidence and mortality numbers, complications etc. I don't think there is need to mention terms like bioenergetics.
Answer: We followed the suggestion of the reviewer
It is unclear what is the purpose and explanation for differentiating between PRRs (pattern-recognition receptors) and non-PRRs.
Answer: we have clarified this point
Regarding the part of the manuscript describing PAMPs and DAMPs as potential biomarkers, the authors 55.3% is good sensitivity for plasma HMGB1 in predicting multiorgan failure in septic pediatric patients, which is a problematic statement. Also, it would be logical to mentioned the main studied non-PAMP/DAMP biomarkers for sepsis so far.
Answer: we have modified the paper following the suggestion of the reviewer
I find the whole part of the manuscript describing treatment strategies problematic. Main treatment options are based on removal of PAMPs and DAMPs which is an oversimplistic view; there is no mention of potentially deleterious consequences of PAMP, DAMP, cytokine removal or neutralization on effective immunologic response and resolution of infection. Likewise, there is no commentary on the effect of “removal technologies” on antibiotic pharmacokinetics. There is even mention of “moderate antibiotic treatment” and septic molecules removal combination resulting in 100% survival of septic mice, which is not further explained. Disregarding this non-comprehensive part of the manuscript, most of the manuscript is written like a chapter from an immunology book, lacking novel, clinically meaningful data.
Answer: we have improved the paper. We hope the reviewer will find the new version more interesting
The manuscript needs to undergo extensive grammatic and syntactic corrections.
Some examples:
- starting a sentence with „Anyway,“, or
- „Cell swelling and plasma membrane rupture, under extreme chemical or physical insults such as the presence of toxins or trauma, characterize it.“ or
- „Human Immunodeficience Virus“ or
- „It is well known that sepsis is a deregulated host response to infection, and in the previous paragraphs, it resulted evident that the first deregulations involve the immune cells.“
Answer: the paper has now been corrected by an English mother tongue
Please find more specific answer in the file attached

Reviewer 2 Report
Comments and Suggestions for Authors
Dear editors, dear colleagues,
the present review of Cicchinelli and coworkers deals with one of the most complex diseaese patterns -sepsis. It highlights the molecular mechanisms contributing to the derailing of the immune response. There are numerous reviews regarding this field but this well written and clear organized review assists the interested readership with tables and figures. Furthermore, it focuses on the janus-faced functions of many DAMPs and PAMPs in the temporal and spatial progress of septical events. Last but not least, it adresses potential targets of diagnostical and therapeutical approach.
I have only two minor questions:
- in the first half of the review TLR8 is given as sensor for ssRNA, in the second half TLR7 is named. Maybe it is advantegous to turn to TLR7/8?
- maybe a figure or table should provide informations about tissue specific expression of TLRs?
Nevertheless, I feel the review should be considered for publication in the present form..
Best regards
Author Response
Dear editors, dear colleagues,
the present review of Cicchinelli and coworkers deals with one of the most complex diseaese patterns -sepsis. It highlights the molecular mechanisms contributing to the derailing of the immune response. There are numerous reviews regarding this field but this well written and clear organized review assists the interested readership with tables and figures. Furthermore, it focuses on the janus-faced functions of many DAMPs and PAMPs in the temporal and spatial progress of septical events. Last but not least, it adresses potential targets of diagnostical and therapeutical approach.
Answer: Thanks for your comment.
I have only two minor questions:
- in the first half of the review TLR8 is given as sensor for ssRNA, in the second half TLR7 is named. Maybe it is advantegous to turn to TLR7/8
Answer: As suggested since both TLR7 and TLR8 bind ssRNA, they have been named together as TLR7/8;
- maybe a figure or table should provide informations about tissue specific expression of TLRs
Answer: We have added a table to provide information about tissue specific expression of TLRs (see Table 3).
Nevertheless, I feel the review should be considered for publication in the present form.
Best regards
Round 2
Reviewer 1 Report
Comments and Suggestions for Authors
The authors have sufficiently responded to my comments and significantly improved the manuscript. My main suggestion is to try to shorten the manuscript (in my opinion is still too long). I do not have any further complaints.
Comments on the Quality of English LanguageMinor editing of English language required
Author Response
We appreciate that – after the revision – you have found that the quality our paper has improved, and it responded to your request.
With regard to the new revision, with particular focus on the first part of the manuscript, we have removed superfluous sentences (mainly addressing the reader to the existing figures and tables for further explanation of some concepts), and ordinated and summarized redundant notions. To further abbreviate the article we have summarized the paragraph regarding the different expression of DAMPs and PAMPs in SOFA-related organ and systems in a new Table (see Table 4).
As a consequence of these changes many references have been renumbered.
A minimal additional English revision has been made.